# Sphingomyelin regulates the transcriptional machinery in nuclear lipid microdomains
Carmela Conte[1,11], Michela Bulfoni[2,11], Federico Fiorani[1], Samuela Cataldi[1], Nicolò Gualandi[2], Ornella Calderini[3], Mercedes Garcia-Gil[4,5], Giorgia Vesca[2], Rita Paroni[6], Michele Dei Cas [6], Cataldo Arcuri[7], Alessandra Mirarchi[7], Tommaso Beccari[1], Toshihide Kobayashi [8,9], Nario Tomishige [8,9], Paola Signorelli[10], Francesco Curcio [2]✉ & Elisabetta Albi [1]✉

Nuclear lipid microdomains rich in sphingomyelin and cholesterol content regulate double-stranded exonuclease-resistant RNA. The study aimed to elucidate the importance of nuclear lipid microdomains in safeguarding nuclear RNA from digestion and to scrutinize all RNA present. Thus, we investigated the impact of sphingomyelinase on nuclear lipid microdomain RNA and conducted RNA extraction, library preparation, and sequencing. Sphingomyelinase treatment makes the RNA susceptible to RNase treatment. Nuclear lipid microdomains exhibit a higher abundance of retained introns, small nuclear RNA, and long intergenic non-coding RNA compared to whole nuclei, with a notable enrichment in miRNA. The high concentration (20%) of miRNAs in nuclear lipid microdomains is justified by the presence of specific nuclear circular RNA as exons circularized with 'retained' introns, referred to as exon-intron circular RNA (EIciRNA) that act as a sponge for miRNAs. Moreover, we demonstrate the presence of ciRNA. The functional analysis indicates that all types of RNase-resistant RNA associated with nuclear lipid microdomains are involved in chromatin organization and brain pathophysiology. In conclusion, nuclear lipid microdomains represent a site of transcription regulation in which circular RNAs, miRNA, and double-stranded mRNA, all resistant to RNase, are stabilized by nuclear sphingomyelin.

The inner nuclear membrane plays a crucial role in various nuclear activities, such as maintaining nuclear integrity, organizing chromatin, and regulating gene expression and transcription[1]. Lipids, particularly sterols and sphingolipids (Sphs), have been identified as key contributors to nuclear function[2]. In contrast to the historical view of lipids as inert structural components of cell membranes, recent research has highlighted the regulatory and signaling roles of Sphs within cell nuclei[3–5]. Nuclear Sphs, including sphingomyelin (SM), ceramide (Cer), sphingosine, and sphingosine-1-phosphate, are believed to modulate processes like DNA replication, repair, chromatin remodeling, and apoptosis[6]. Notably, SM has been

shown to regulate transcription factors, essential proteins governing gene transcription[7]. Evidence suggests that nuclear SM metabolism can function independently of its metabolism in other cellular compartments, with neutral sphingomyelinase (nSMase), an enzyme present in the nucleus, playing a role in hydrolyzing SM to Cer[8].

The distinct metabolic pathway and its potential modulation of nuclear processes are active areas of research, emphasizing its unique role in cellular regulation[4].

Intriguingly, a transmission electron microscopy analysis on purified nuclei revealed that SM predominantly localizes within the perichromatin

[1]Department of Pharmaceutical Sciences, University of Perugia, 06126 Perugia, Italy. [2]Department of Medicine (DAME), University of Udine, 33100 Udine, Italy. [3]Institute of Biosciences and Bioresources, National Research Council (CNR), 06128 Perugia, Italy. [4]Department of Biology, University of Pisa, 56127 Pisa, Italy. [5]Department of Biology, Interdepartmental Research Center Nutrafood "Nutraceuticals and Food for Health", University of Pisa, 56127 Pisa, Italy. [6]Department of Health Sciences, University of Milan, 20142 Milan, Italy. [7]Department of Medicine and Surgery, University of Perugia, 06126 Perugia, Italy. [8]UMR 7021 CNRS, Université de Strasbourg, Strasbourg, 67401 Illkirch, France. [9]Cellular Informatics Laboratory, RIKEN, Wako, 351-0198 Saitama, Japan. [10]Biochemistry Laboratory, IRCCS Policlinico San Donato, 20097 Milan, Italy. [11]These authors contributed equally: Carmela Conte, Michela Bulfoni. ✉e-mail: francesco.curcio@uniud.it; elisabetta.albi@unipg.it

region, housing sites of pre-mRNA synthesis and processing[9]. RNA molecules, typically single-stranded, can form short double helices that are resistant to RNase treatment[10].

Double-stranded RNA (dsRNA) in the nuclear context is present within native ribonucleoprotein complexes[11] and associates with a small protein, constituting a domain crucial for nucleocytoplasmic trafficking[12]. Additionally, nuclear dsRNA plays a role in DNA repair processes[13].

A small fraction of active chromatin, containing RNase-resistant RNA anchored to the inner nuclear membrane, was previously purified and termed the "intranuclear complex"[14]. This complex was found to correspond precisely to a subnuclear fraction obtained by purifying nuclear lipid microdomains (NLMs) rich in SM and cholesterol (Chol) content[15]. NLMs, isolated after removing the external nuclear membrane and digesting nuclei with DNase and RNase, were found to contain a small amount of DNA (0.3% of total DNA) and a fraction of RNase-resistant RNA (10–20% of total RNA), indicating that NLMs are inner nuclear membrane fractions with attached RNase-resistant RNA[15]. Interestingly, it is the SM that protects the RNA from the action of Rnase[16]. Microinjection in cell nucleus of enzymatically active nSMase into living cells resulted in the rapid degradation of intranuclear structures[9]. SM was subsequently shown to interact with Chol to protect RNA from RNase digestion[14]. Transcription factors within NLMs contributed to the regulation of the transcription process[17,18]. It is now understood that the transcriptional machinery is influenced by various types of RNA, categorized into protein-coding and noncoding RNA. CircRNAs, a type of noncoding RNA, are covalently closed, single-stranded molecules devoid of 5' to 3' polarity and polyadenylated tails. They consist of sequences from both exons and introns, displaying resistance to exonuclease digestion[14]. CircRNAs, generated through non-canonical splicing, play crucial roles in cellular pathophysiology, acting as sponges for miRNAs[19,20]. Unlike linear RNA, the transport of circRNAs to the cytosol is not linked to nuclear envelope breakdown during mitosis, instead involving a specific transport mechanism[21]. In the nucleus, circRNAs exist as exon-intron circRNAs (EIciRNAs), with retained introns between exons[22]. CircRNAs regulate transcription initiation and elongation, recruit epigenetic modifiers, and act as miRNA sponges, influencing posttranscriptional regulation in the cytosol[23]. Another specific nuclear circRNA with resistance to debranching and degradation is ciRNA, consisting of only introns that play a role in transcriptional regulation of its parental genes[24,25].

Originating from endogenous nuclear DNA loci, the miRNAs exit the nucleus and undergo maturation in the cytoplasm where they interact with proteins of the Argonaute family and become part of the RNA-induced silencing complex (RISC)[26]. Within the cytosol, miRNAs play crucial roles in regulating posttranscriptional processes[26]. The mature miRNA-coupled RISC is then transported from the cytoplasm back into the nucleus by specific transporter proteins, including importin-8, importin-α/β, and Chromosomal Maintenance 1 (CRM1)[27–29]. Inside the nucleus, miRNAs actively participate in regulating the transcription process[30].

As of now, the composition of RNase-resistant RNA associated with NLM has not been elucidated. The NLM has been demonstrated in liver and hepatocytes[9,14,15], T24 human bladder carcinoma and V79 Chinese hamster cells[9], SUP-T1 human lymphoblastic lymphoma[31], and HN9.10e embryonic hippocampal cells[18]. In this work, we studied the RNase-resistant RNA present in NLM purified from HN9.10e cells with the aim of evaluating whether there were types of RNA correlated with the functioning of embryonic and/or stem cells of hippocampus, given the general interest in scientific research and our interest in the pathophysiology of this brain structure. Thus, this study represents a pioneering exploration into the characteristics of RNAs protected from RNase degradation due to the presence of SM in NLMs of HN9.10e cells.

## Results
### Purity of nuclear lipid microdomains

Highly purified nuclei were used to prepare NLMs. To test the purity of nuclei and NLMs, the immunoblot analysis with giantin antibody, as a marker protein for Golgi membrane was performed[15]. Figure 1 showed that the band of giantin, corresponding to the apparent molecular weight of 367 kDa, was present in cells; a very low amount was present in nuclei and it was completely absent NLMs. Then, the STAT3 and lamin B were tested as

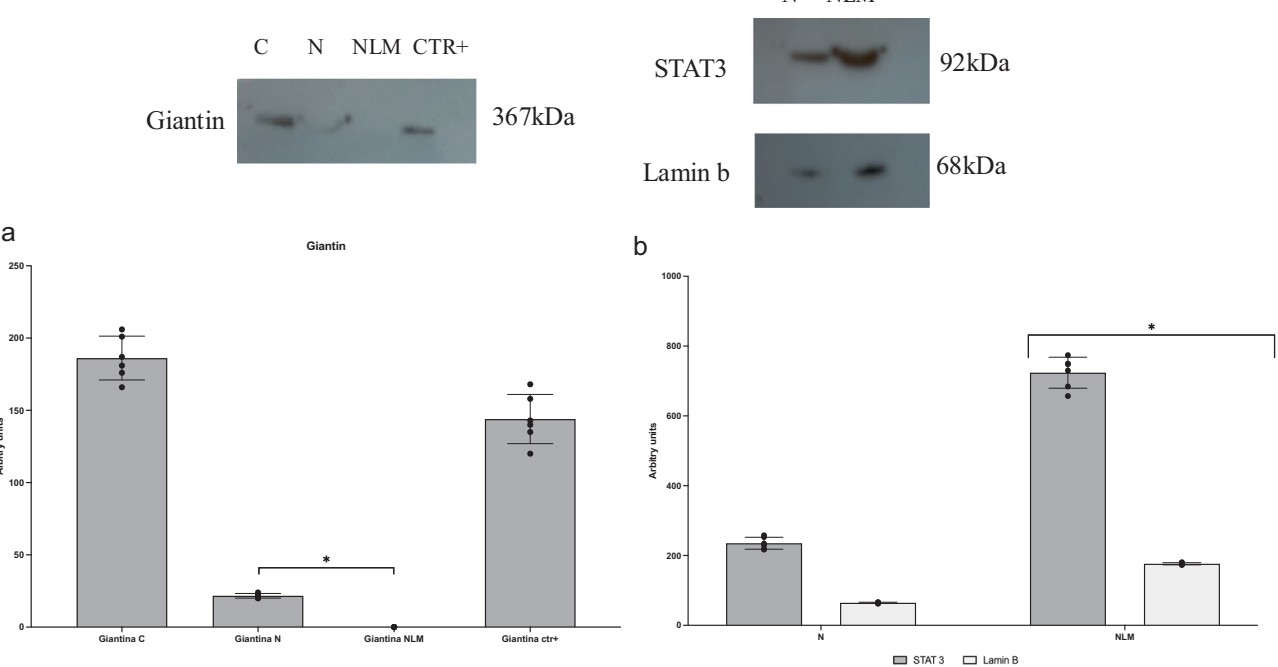

**Fig. 1 | Western blotting of marker proteins in purified nuclei and nuclear lipid microdomains.** Western blotting analysis of: **a** giantin, as marker protein of Golgi membrane in HN9.10e cells (C), purified nuclei (N), nuclear lipid microdomains (NLM) and, SUPT1 cells as positive control (CTR+); **b** STAT3 and lamin b, as markers of NLMs in purified nuclei (N) and nuclear lipid microdomains (NLM).

Above images of western blotting and below quantification of area density by Chemidoc Imagequant LAS500 by the specific IQ program. Data are expressed as mean ± SD of three independent experiments performed in duplicate. For the significance, *t* test was used; *$p < 0.01$, **a** versus C and in **b** versus N.

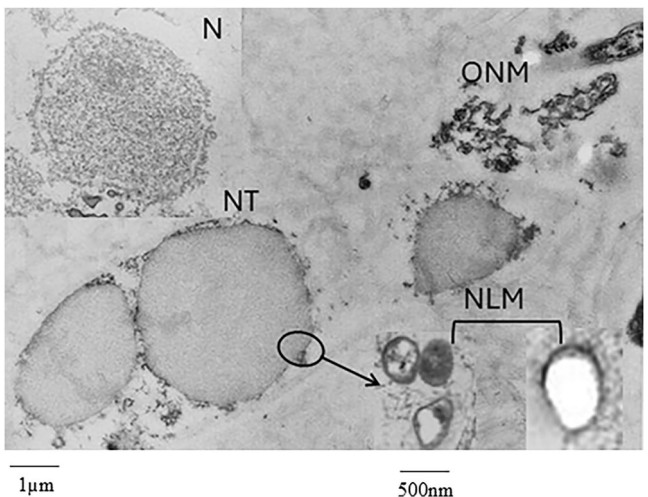

**Fig. 2 | Ultrastructural analysis of nuclei and nuclear lipid microdomains (NLMs).** The figure represents a montage of electron microscopy images of purified nuclei (N), nuclei depleted of outer nuclear membrane by Triton-X100 treatment (NT) with residues of outer nuclear membrane (ONM), purified nuclear lipid microdomain (NLM). N and NT have an average diameter in the range of 6–7 μm. N appears well purified with a nuclear membrane well defined and complete and NT have lost their ONM that appears separate (12,000×). After washing and centrifugation of NT to remove the ONM, the NLMs are purified and appear as a population of closed, spherical or ovoid vesicle-like structures with an average diameter in the range of 300–600 nm (on the left image 65,000×; on the right image 200,000×).

marker of NLMs[15,17]. The results showed a higher STAT3 and lamin B protein content in NLMs, according to previously obtained results in NLMs purified from liver and hepatoma cells[15,17] (Fig. 1, Supplementary Fig. 1).

The purity of the nuclei, nuclei treated with Triton X-100, and NLMs was controlled by electron microscopy analysis (Fig. 2). The nuclei were highly purified with an average diameter in the range of 6–7 μm. After treatment of the nuclei with sucrose solution containing Triton X-100, used to remove the outer nuclear membrane, preserving the inner nuclear membrane, the analysis of the pellet shows the nuclei deprived of the outer nuclear membrane and residues of the outer nuclear membranes. After washing of nuclei treated with Triton X-100 and purification of NLMs, the pellet is characterized by a homogenous population of closed, spherical, or ovoid vesicle-like structures with an average diameter in the range of 300–600 nm, as previously reported for the NLMs purified from hepatocytes[15].

In hepatocytes, the composition of NLMs analyzed by thin layer chromatography has been shown to be characterized by a strong interaction SM-Chol[14]. Moreover, SM and Chol have been proposed to drive diverse actions in the cell nucleus, especially in the nervous system[18]. Furthermore, it is not known the total lipid composition of NLM. Thus, we tried to expand the study by carrying out untargeted lipidomics with the aim of identifying the differences in the lipid components of whole nuclei and NLMs of embryonic hippocampal cells (HN9.10 cell line). The content of proteins in nuclei purified from HN9.10 cells and in isolated NLMs was $108.9 \pm 2.30$ μg/$10^6$ cells and $1.69 \pm 0.03$ μg/$10^6$ cells, respectively. Based on lipidomic analysis, different lipids were identified in both purified nuclei and isolated NLMs. In particular, neutral lipids as Chol, cholesterol ester (CE), free fatty acid (FA), triacylglicerol (TG), triacylglycerol estolides (TG_EST), diacylglycerol (DG), ether-linked diacylglycerol (EtherDG), monoacylglycerol (MG), n-acyl-ethanolamina (NAE), ganglioside GM 3 (GM3). Moreover, phospholipids (phosphatidic acid (PA), phosphatidylglycerol (PG), lysophosphatidylglycerol (LPG), ether-linked phophatidylglycerol (EtherPG), ether-linked lysophosphatidylglycerol (EtherLPG), phosphatidylcholine, PC; lysophosphatidylcholine, LPC; ether-

linked phosphatidylcholine, EtherPC; ether-linked lysophosphatidycholine, EtherLPC; phosphatidylethanolamine, PE; lysophosphatidylethanolamine, LPE; ether-linked phosphatidyethanolamine, EtherPE; ether-linked lyso-phosphatidyethanolamine, EtherLPE; n-acyl-lysophosphatidyethanolamine, LNAPE; phosphatidyinositol, PI; lysophosphatidylinositol, LPI; ether-linked phosphatidylinositol, EtherPI; phosphatidyserine, PS; lysopho-sphatidylserine, LPS; ether-linked phosphatidylinoserine, EtherPS) and sphingolipids (sphingomyelin, SM; ceramide, Cer; dihydroceramide, DhCer; ceramide alpha-hydroxy fatty acid-phytosphingosine, Cer AP; ceramide esterified omega-hydroxy fatty acid-sphingosine, Cer EOS; ceramide hydroxy fatty acid dihydrosphingosine, Cer HDS; ceramide hydroxy fatty acid sphingosine, Cer HS; hexosylceramide, HexCer; hexosylceramide hydroxy fatty acid sphingosine, HexCer HS; lactosylceramide, LacCer; ceramide phosphatidylinositol, PI Cer) were identified in both nuclei and NLMs. By comparing the main lipid classes, a higher percentage of polar lipids was evident in the NLMs compared to the nuclei (Fig. 3a). Moreover, NLMs were richer than nuclei in total Chol, SM and Cer content (Fig. 3b). If the value of the arbitrary units/mg protein is computed considering the molecular weight, it would appear that in the nuclei the Chol was equal to $3.84 \pm 0.01$ μmol/mg protein and the SM was equal to $34.77 + 0.53$ μmol/mg protein while in the NM the Chol was equal at $42 + 0.27$ μmol/mg protein and SM was equal to $50.12 + 0.18$ μmol/mg protein. The similarity of molecular content of Chol and SM in NLMs is according to Scassellati et al.[9] and Rossi et al.[14]

## Role of sphingomyelin and cholesterol in the nuclear lipid microdomains from HN9.10e cells

The role of SM-Chol in RNAse-resistant RNA of hepatocyte nuclei was well established[14]. To study the nuclear localization of SM and the SM-Chol microdomains in HN9.10 cells we used fluorescent probes (Fig. 4a–d). The results showed that SM, identified with the Lys probe, was diffusely present in the cell membrane and in proximity to the nucleus, counterstained with DAPI. The SM-Chol microdomains, highlighted with the enhanced green fluorescent protein-nakanori (NAK) probe, were present in specific areas of the cell membrane and in the nuclear periphery, according to Scassellati et al.[9]

In the HN9.10 cell nuclei, the DNA and RNA content corresponded to $9.08 \pm 0.07$ and $1.8 \pm 0.21$ μg/$10^6$ cells, respectively. In the NM, the level of DNA and RNA was $0.032 \pm 0.004$, and $0.16 \pm 0.02$ μg/$10^6$ cells, respectively, indicating that about 1/10 of the nuclear RNA was resistant to the RNase treatment performed during the NLM extraction. We further implicated SM and Chol, located in NLMs in the maintenance of the nuclear RNase-resistant RNA, as occurs in hepatic cells[9,14]. Thus, we performed experiments incubating HN9.10e cells with increasing concentrations of SM-Chol, as reported in the "Methods" section, and the incorporation of the two lipids in NLM was analyzed. As expected, the content of SM and Chol increased (Fig. 5a). To better understand which types of SM NLMs were enriched with, we carried out a target UFLC MS/MS by treating the cells with the highest concentration of SM-Chol (20 μM). Specifically, SM with a 16-carbon saturated fatty acid (16:0SM), SM with an 18-carbon mono-unsaturated fatty acid (18:1SM), and SM with a 24-carbon saturated fatty acid (24:0SM) were analyzed. Thus, we treated the cells with a pool of SM in equimolecular concentration with CHO, as reported in the methods, and studied the most common SM species by using external calibrators for each species (16:0SM, 18:1SM, 24:0 SM in a ratio of 1/1/1) added to the purified NLM samples before lipid extraction. We observed that exogenous SM-Chol induced an enrichment in saturated 16:0 SM and 24:0 SM species without significant variation of monounsaturated 18:1 SM (Fig. 5b).

At the same time, the NLM resulted richer in RNAse-resistant RNA (Fig. 5c). The action of SM-Chol on the stabilization of the nuclear RNAse-resistant RNA was supported by treating the NLM with SMase, capable of degrading SM and releasing Chol. The subsequent treatment with RNase in both the control NLMs and the nSMase-treated NLMs showed that in the control the RNA content remained unchanged, while in the experimental NLMs, the RNA was present only in traces (Fig. 5d). For comparison, the

**Fig. 3 | Untarget lipidomic in nuclei (N) and nuclear lipid domains (NLM). a** the values are expressed as percentage of total neutral and polar lipids; **b** analysis of cholesterol and sphingolipid peaks, the values are expressed as arbitrary units/mg proteins. Chol Cholesterol, SM sphingomyelin, Cer ceramide, DhCer dihydroceramide, Cer AP ceramide alpha-hydroxy fatty acid-phytosphingosine, Cer EOS ceramide esterified omega-hydroxy fatty acid-sphingosine, Cer HDS ceramide hydroxy fatty acid dihydrosphingosine, Cer HS ceramide hydroxy fatty acid sphingosine, HexCer hexosylceramide, HexCer HS hexosylceramide hydroxy fatty acid sphingosine LacCerlactosylceramide. For both **a** and **b**, the data are expressed as the mean ± SD of 3 independent experiments. For the significance, *t* test was used; *$p < 0.05$ versus N.

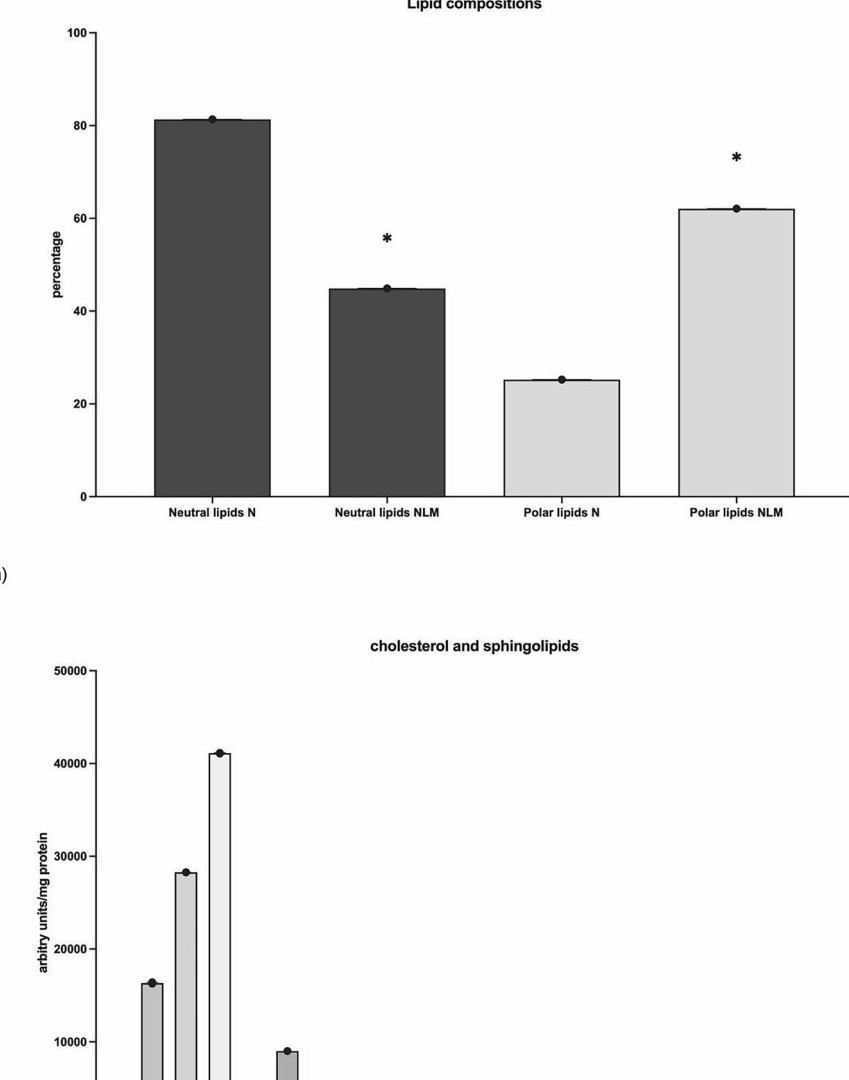

experiment was repeated with PC-PLC treatment, capable of degrading PC, without obtaining any significant variation compared to the control sample (Fig. 6b). Thus, the SM-Chol complex played a role in retaining RNA and protecting it from RNase degradation.

In light of our previously published results in which SM was essential for the differentiation of HN910e cells with a change of the soma characteristics and with the appearance of neurites[32], we hypothesized a differentiation effect induced by 20 µM SM-Chol. As shown in Fig. 6a, the control sample was characterized by a very low percentage of cells with short neurites. 20 µM SM-Chol induced a reduction in cell growth and cell differentiation, characterized by modification of the soma in a large number of cells. The length of neurites was significantly increased; the cells acquired neuron and astrocyte characteristics (Fig. 6b, c). It has been confirmed by labeling the cells with GFAP as a marker of astrocytes and βIII-tubulin as a marker of neurons (Fig. 6d). The results showed that the control sample had low levels of GFAP expression (in red) and absence of βIII-tubulin expression (in green), characteristic of stem cells. The immunolabelling was counterstained with DAPI (in blue). Experimental cells showing strong βIII-tubulin immunolabeling and weak GFAP expression indicated cells in the process of differentiating toward a unipolar cellular neuronal phenotype, but still with precursor cell properties.

## Characteristics of RNA protected against RNase action by sphingomyelin and cholesterol in nuclear lipid microdomains

With the aim of establishing which types of RNA were resistant to the action of RNase in NLMs, total RNA was extracted from NLMs and analyzed by comparing its composition with that of the RNA present in the whole nuclei. The results showed that the main RNAs present in the whole nuclei were 60% protein-coding RNA, 8% miscellaneous RNA (miscRNA), 5% small nuclear RNA (snRNA), 3% nonsense-mediated decay RNA, and 1% long intergenic non-coding (lincRNA). In the NLMs, the main RNAs were 20% protein-coding RNA, 20% miRNA, 16% miscRNA, 8% retained intron, 5% snRNA, 3% lincRNA, 3% nonsense-mediated decay RNA, and 3% small nucleolar RNA (snoRNA). Specifically, protein-coding RNA was substantially more abundant in whole nuclei *vs* NLMs ($p = 0.0007$) while miRNA levels were markedly higher in NLMs ($p = 0.0022$). These findings suggest that the RNA composition differs considerably between N and NLMs, with certain RNA classes, such as protein coding RNA and miRNAs

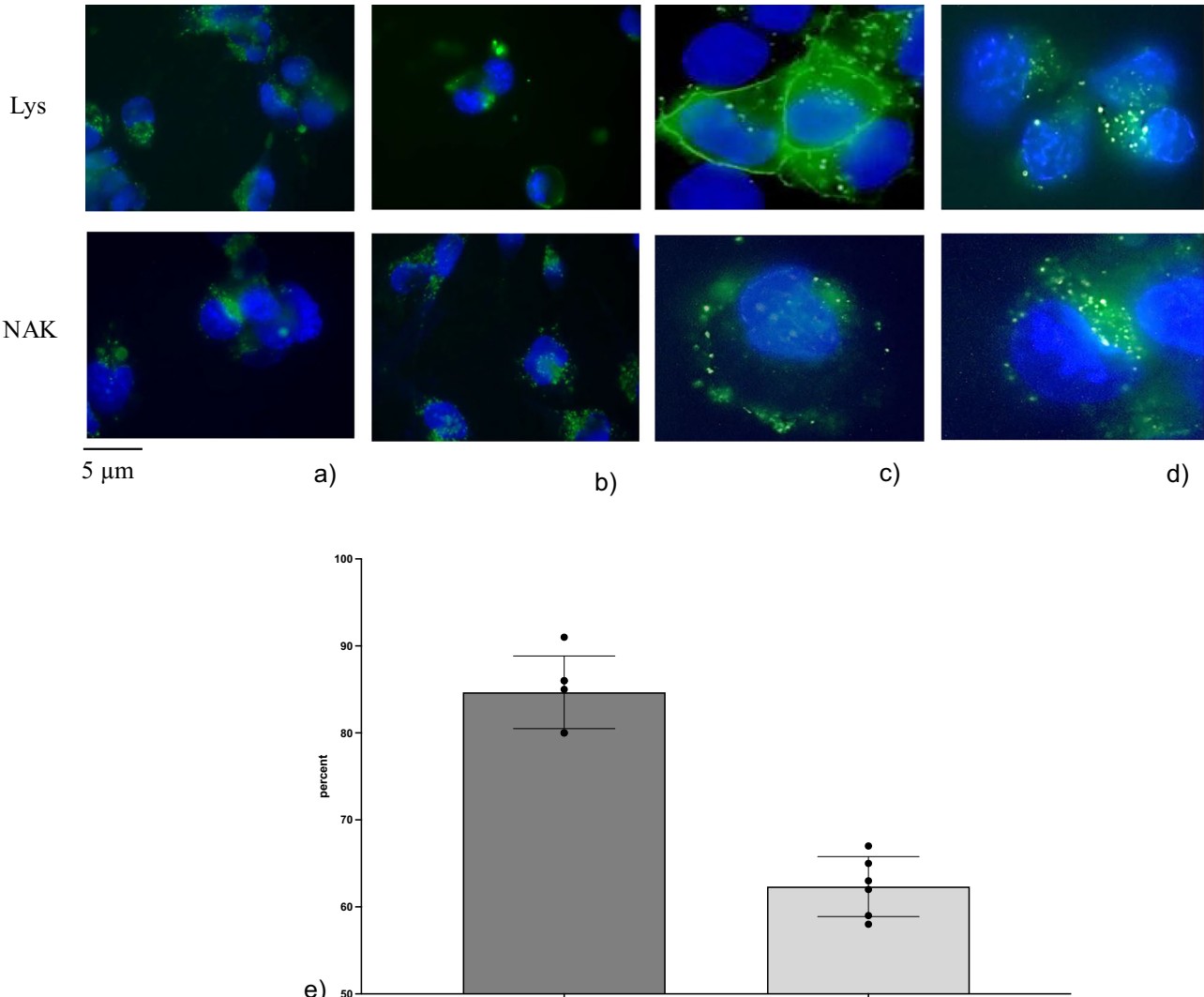

**Fig. 4 | Localization of sphingomyelin with EGFP-NT-Lys probe (Lys) and sphingomyelin-cholesterol microdomains with EGFP-nakanori probe (NAK).** **a**, **b** images with 40 magnification (bar 5 μm), **c**, **d** images with 100 magnification (bar 5 μm), **e** percentage of positive cells, the data represent the mean ± SD of 4 independent experiments.

being preferentially enriched in one compartment over the other. All other RNA biotypes, present in lower amounts, are fully reported in Fig. 7.

In detail, the chart shows the distribution of biotypes in the nuclei and NLM samples, represented through a stacked bar plot. Each color corresponds to a different RNA biotype, as indicated in the legend below the Figure. Among the various biotypes represented, green identifies antisense RNA, blue corresponds to bidirectional promoter long non-coding RNA, while orange indicates protein-coding sequences. Purple represents processed transcripts, yellow denotes ribosomal RNA, and red represents microRNA.

Additionally, brown is associated with miscellaneous RNAs, pink with polymorphic pseudogenes, while light blue identifies sense-overlapping sequences. Dark purple represents snRNA, dark green indicates ribozymes, and gray corresponds to macro lncRNA sequences. The legend follows the same color order used in the bars to facilitate the interpretation of the chart.

Therefore, the NMs were less rich in mRNA and snoRNA compared to the nuclei but were richer in miscRNA, retained intron, snRNA, and lincRNA, and indeed abundant in miRNA. Given that protein-coding RNAs, miRNAs, and miscRNAs (which may also include circRNAs) were the most differentially expressed and prominent RNA biotypes emerging from our analysis, we decided to further investigate them.

## mRNA in NLMs and nuclei

First, differentially expressed mRNAs were analyzed in NLMs and nuclei. By comparing the principal components analysis, we were able to confirm the good reproducibility of all biological replicas (Supplementary Fig. 2). mRNA-seq (abs(logFC) ≥ 0.6, $q$-value ≤ 0.05) analysis was performed to identify DEmRNAs occurring between NM and nuclei (Supplementary Fig. 3). Among all the mRNAs profiled through the RNA-seq experiment ($n = 30942$), a total of 20 mRNAs resulted differentially concentrated in NLMs respect to nuclei in a statistically significant manner. NLMs exhibited a significant higher concentration of 14 mRNAs in comparison with nuclei and 6 mRNA were present significantly lower than nuclei (Fig. 8). The mRNAs present in high concentration in the NLMs were Kansl1 (FDR value:6.611E−68), Ehmt2 (FDR value:4.37E−09), Timeless (FDR value:5.321E−53), Inf2 (FDR value:8.481E−126), Rbm28 (FDR value:1.265E−08), Prpf6 (FDR value:3.386E−14), Chaf1a (FDR value:1.6421E−108), Mtdh (FDR value:3.347E−12), Fam114a1 (FDR value:3.753E-48), Daam2 (FDR value:1.573E−11), Ankrd11 (FDR value:3.928E−48), Spout1 (FDR value: 1.526E−19), Nicn1 (FDR value:1.958E−13), Noc2l (FDR value:2.977E−1). Data on three representative transcripts with high differential expression between nuclei and NLMs were validated using an orthogonal reference method, RT-qPCR, in

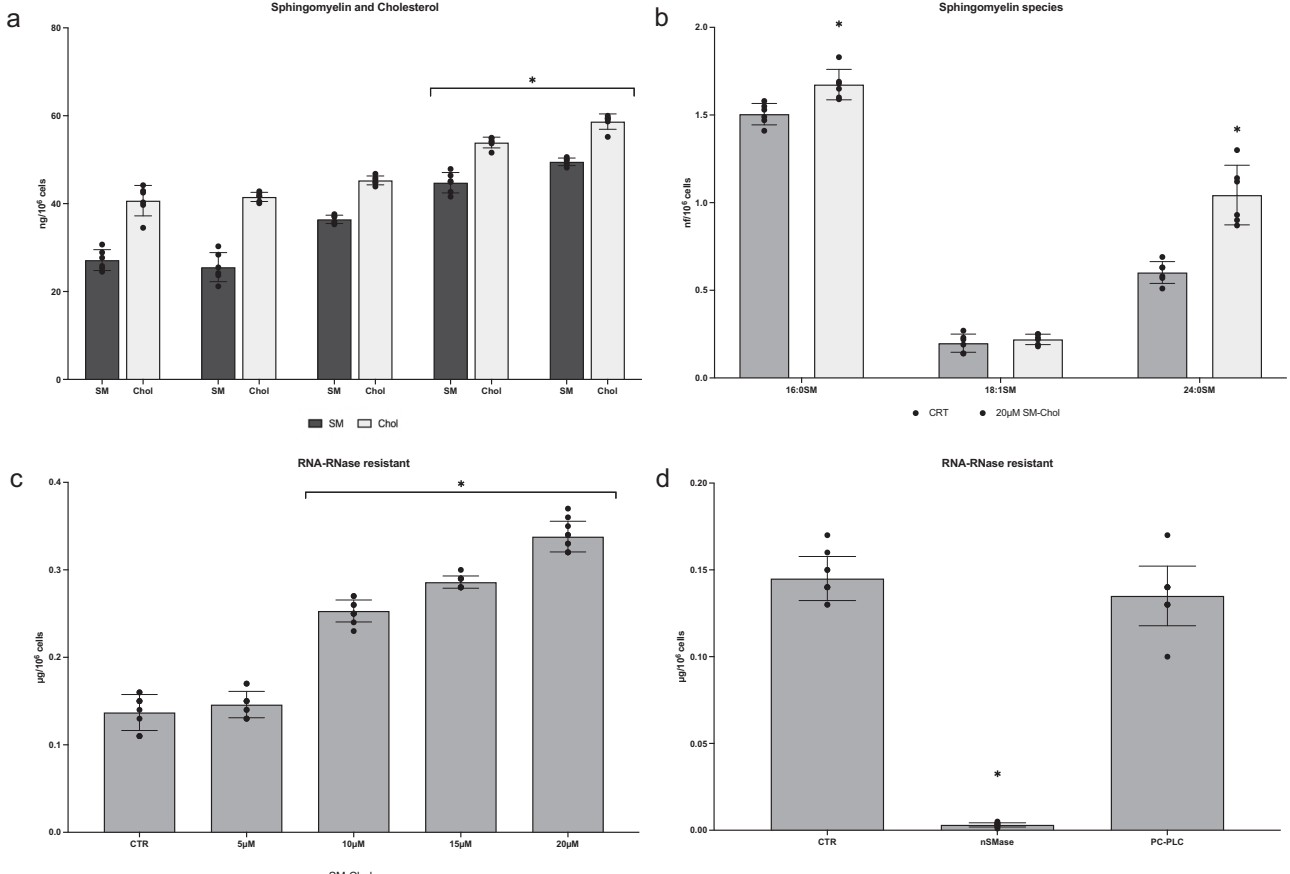

**Fig. 5 | Effect of HN9.10e cell treatment with a pool of sphingomyelin (SM) and cholesterol (Chol) on nuclear lipid microdomains (NLMs) and on RNA-resistant RNase. a** changes in total SM and CHO content in NLMs after incubating cells with increasing concentrations of a pool of SM and CHO, reported in the abscissae; **b** changes of the most common SM species in NLM purified after incubation of the cells with 20 μM SM-CHO. Specifically, SM with a 16-carbon saturated fatty acid (16:0SM), SM with an 18-carbon monounsaturated fatty acid (18:1SM), and SM with a 24-carbon saturated fatty acid (24:0SM) were analyzed. Lipids were quantified as described in "Methods" section; **c** analysis of RNase-resistant RNA in nuclear lipid microdomains in the presence of increasing equimolecular concentration of sphingomyelin (SM) and cholesterol (Chol); **d** analysis of RNase-resistant RNA in nuclear lipid microdomains after neutral sphingomyelinase (nSMase) and phosphatidylcholine-dependent phospholipase C (PC-PLC) treatment. Data are expressed as mean ± SD of 3 independent experiments performed in duplicate for **a** and **b** and as mean ± SD of 5 independent experiments performed in duplicate for **c** and **d**. For the significance, *t* test was used; *$p < 0.05$ versus control, CTR.

independent samples (nuclei $n = 3$; NLMs $n = 4$). Specifically, Ehmt2, Chaf1a, and Dnm2 confirmed the observed up-regulated trends, with fold changes of 216, 5571, and 1382, respectively.

The functional analysis of genes highly concentrated in the NLMs compared to the nuclei, which are targeted by miRNAs highly concentrated in the NLMs compared to the nuclei according to associations from TarBase (citazione: https://pmc.ncbi.nlm.nih.gov/articles/PMC1370898/), clearly indicates their involvement in chromatin remodeling and organization, as well as in the development of the nervous system (Supplementary Fig. 4). Notably, 224 genes were associated with the protein modification process, while 159 genes were involved in neurogenesis, and 137 genes played a role in the generation of neurons. Additionally, 132 genes were linked to neuron differentiation, and 109 genes were involved in the positive regulation of protein metabolic processes. Other key processes included positive regulation of intracellular signal transduction (88 genes), neuron projection morphogenesis (69 genes), protein-DNA complex organization (70 genes), and chromatin organization (68 genes). Furthermore, brain development (65 genes), chromatin remodeling (65 genes), regulation of DNA metabolic process (49 genes), and regulation of neuron projection development (49 genes) were also significantly represented. Additional enriched categories included Neuronal System (33 genes), regulation of neuron apoptotic process (28 genes), heterochromatin organization (19 genes), regulation of postsynaptic membrane neurotransmitter receptor levels (18

genes), and regulation of transcription regulatory region DNA binding (9 genes).

### miRNAs in NLMs and nuclei

Then, differentially expressed miRNAs were analyzed in NLMs and nuclei. The specific miRNA-seq protocol employed is designed to capture small RNA molecules, primarily mature miRNAs. Additionally, the miRBase database is optimized to map sequencing reads specifically to mature miRNAs. Consequently, pre-miRNAs were not detected in our miRNA-seq analysis.

By comparing the principal components analysis, we were able to confirm the good reproducibility of all biological replicas (Supplementary Fig. 5). RNA-seq ($\text{abs}(\log FC) \geq 0.6$, $q$-value $\leq 0.05$) analysis was performed to identify DEmiRNAs occurring between NLMs and nuclei (Supplementary Fig. 6). Among all the miRNAs profiled through the RNA-seq experiment ($n = 1961$), a total of 22 miRNAs resulted in different concentrations in a statistically significant manner, including 14 low-concentrated and 8 high-concentrated miRNAs (Fig. 9). The miRNAs present in high concentration in the NLMs were: 151-3p, 27a-3p, 1198-5p, 345-3p, 669c-5p, 32-5p, 328-3p, 486a-5p. The association between these miRNAs and various diseases is reported in Supplementary Table 1, highlighting their strong correlation with neuro-psychological disorders such as chronic epilepsy, amyotrophic lateral sclerosis, and other psychological

**Fig. 6 | Differentiation of HN9.10e cells with 20 μM SM-Chol.** Cell morphology of untreated (control, **a**) and treated samples (experimental, **b**, **c**), 40x magnification. **d** immunofluorescence with GFAP (red) as marker of astrocytes and βIII-tubulin (green) as marker of neurons. The immunolabelling is counterstained with DAPI (in blue), 100x magnification.

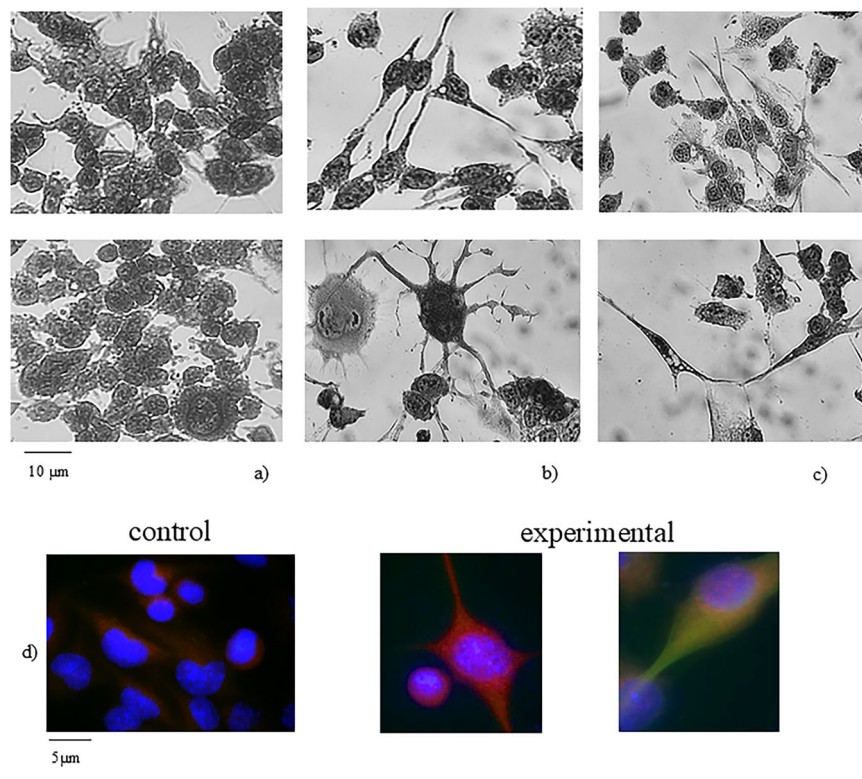

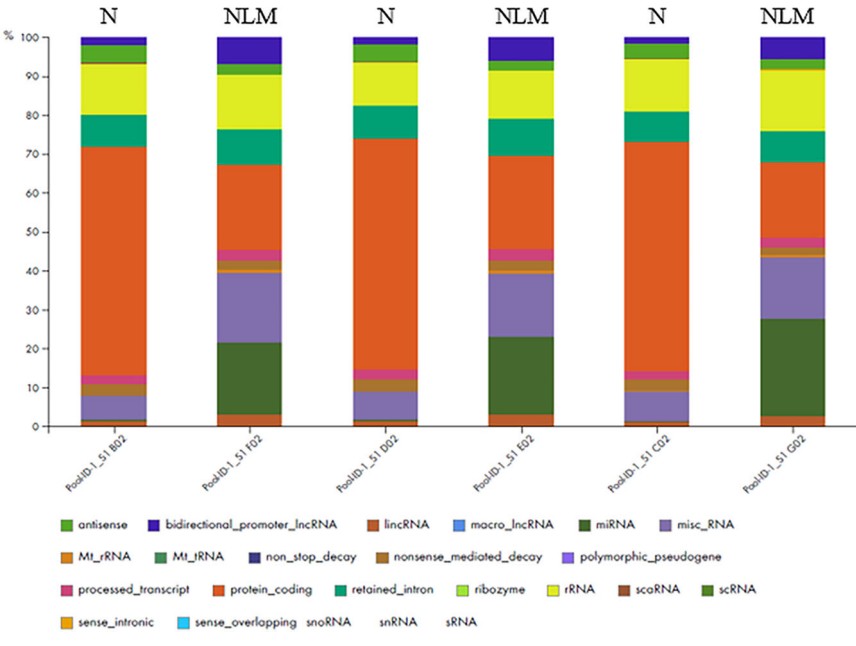

**Fig. 7 | BIOTYPE Chart.** Different composition in RNA between nuclei (N) and nuclear lipid micro-domains (NLM).

conditions. These correlations were predicted using Ingenuity Pathway Analysis (IPA), which integrates extensive knowledge from the literature to explore the biological impact of miRNA expression changes. IPA facilitates this analysis both by leveraging its curated database on miRNA functions and by examining the mRNA targets they regulate, providing deeper insights into their potential roles in disease mechanisms.

Our findings revealed a notable concentration of miRNAs within NLMs. Considering that circRNAs in the nucleus adopt the form of exon-intron circRNAs (EIciRNAs), characterized by retained introns between exons that function as miRNA sponges[20], our study aimed to explore the

potential presence of EIciRNAs within NLMs. The results revealed 18 exceptionally significant EIciRNAs, each comprised of varying numbers of exons (1–5) and introns belonging to ZFX, STRN3, TRUB1, CCNT2, PKP2, CTNNA1, CYREN, DEPDC5, SLC24A2, CSTF3, ACACA, ASCC1, KMT2D, ERCC6, PPP1R9A, PPFIBP1, ANP32P, and CCDC171 genes. Furthermore, NLMs harbored 3 ciRNA species exclusively consisting of introns[21,22] belonging to RECQL5, IRAK1 AND KMT2D genes (Supplementary Table 2). The presence of in NLMs of circRNAs Slc24A2, Ccnt2, and Deodc5 was confirmed by RT-qPCR.

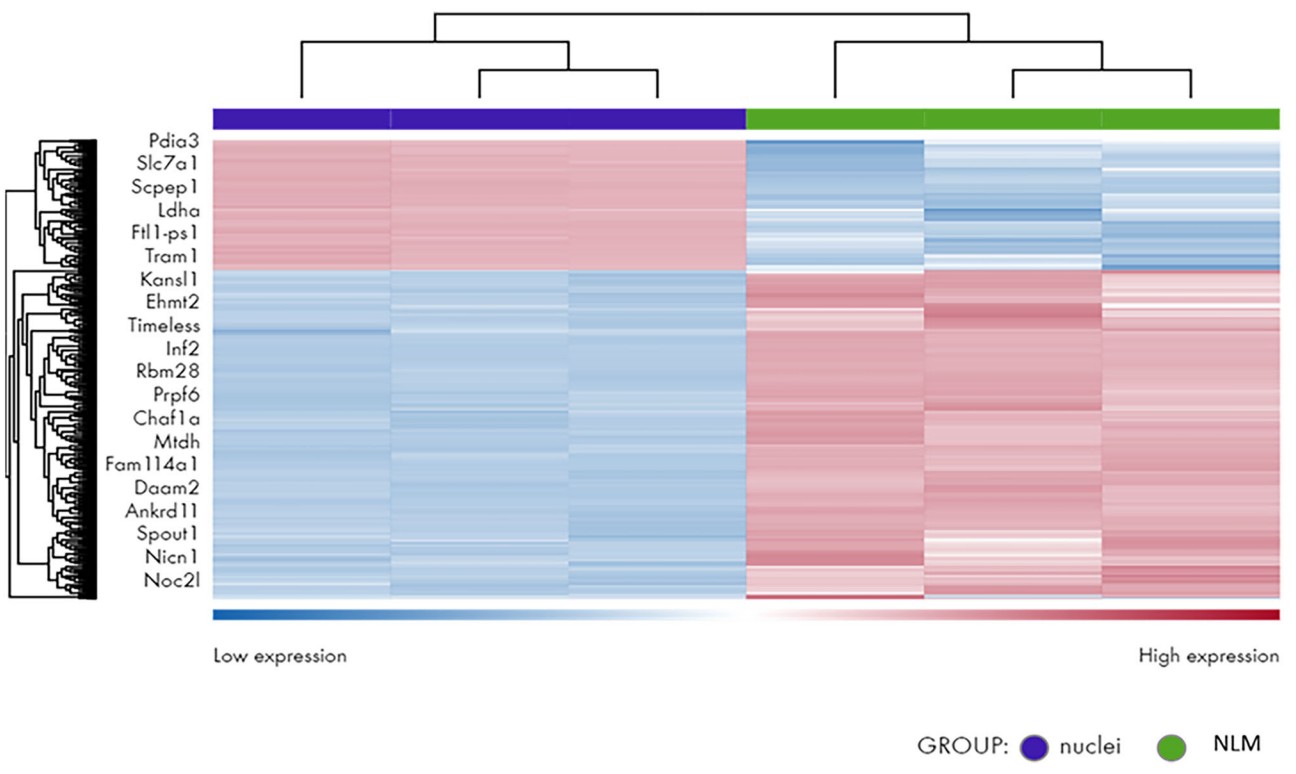

**Fig. 8 | mRNA profiling of nuclear lipid microdomains (NLM) compared to nuclei.** Heatmap showing the statistically significant differentially expressed mRNAs in NLMs respect to the nuclei. Globally, twenty transcripts were differentially present, $n = 6$ low-concentrated, $n = 14$ highly-concentrated in the NLMs in comparison with nuclei (abs(logFC) ≥ 0.6, $q$-value ≤ 0.05). Hierarchical clustering of transcripts and samples using the Euclidean distance and the complete agglomeration method; expression data was VST-transformed, scaled and centered. When not available, the mRBase accession number was replaced by the ENSEMBL GeneID. The heatmap was generated using the DESeq2 R package (v.1.0.12).

## Discussion

Nuclear SM is located in the nucleolus, chromatin, nuclear matrix and inner nuclear membrane, where it binds Chol to form SM-Chol microdomains named NLMs that regulate mRNA transcription[9,14]. The results from this study reveal a key role for NLMs in the protection of specific mRNA and miRNA. Here, we demonstrate that the incubation of the cells with SM-Chol increases the amount of RNase-resistant RNA in NLMs of embryonic hippocampal cells. In exploring the role of SM-Chol, we determined whether the RNA RNase-resistant was decreased by nSMase treatment, an enzyme that degrades SM, freeing Chol, and thereby disorganizing NLMs. Importantly, the results demonstrate that nSMase reverses the effect of SM-Chol and therefore NLMs are necessary to protect the RNA from RNase action, in agreement with Scassellati et al.[9] and Rossi et al.[14]

Our data demonstrated that the reduction of SM and consequently of Chol is responsible for the strong decrease in RNase-resistant RNA, identified in the literature as double-stranded RNA[11–13] and circRNA[14]. Previous studies have shown that intranuclear injection of nSMase leads to disorganization of RNA near the inner nuclear membrane[9]. At this point, it was important to understand which RNAs were protected by the SM-Chol in NLMs. The results show that only 20mRNAs are differentially expressed in the nuclei and NLMs, including 14 highly concentrated mRNAs and 6 low-concentrated mRNA in NLMs. Previous research demonstrated that NLMs acted as rafts for anchoring of transcriptionally active chromatin[9,17]. So we decided to also analyze the mRNA. The functional study highlights the role of these genes in neuronal function, as reported in the results. Interestingly, multiple studies have suggested that the highly concentrated mRNAs in NLMs are essential for nuclear processes and are involved in brain pathology when altered. KANSL1, known to influence the chromatin status as regulator of histone acetylation[33], is linked to the intellectual disability Koolen-de Vries syndrome[34]; EHMT2 that regulates histone methylation is considered a target in pediatric and adult brain tumors[35]; TIMELESS protects replication in difficult-to-replicate areas[36]; INF2 influencing actin polymerization and depolymerization is impaired in glioblastoma[37]; RBM28 inhibits the transcriptional activity of oncosuppressor p53[38]; PRPF6 results mutated in retinitis pigmentosa[39]; CHAF1A influences the chromatin assembly[40,41]; MTDH is involved in glioblastoma[42]; FAM114A1 or NOXP20 is detected in teratocarcinoma-derived neuroectodermic precursor cells[43]; DAAM2 is associated to glioma[44]; ANKRD1 is associated to epilepsy and intellectual disability[45,46]; NOC2L inhibits p53 via inhibition of histone acetyltransferase[47]. Overall, this functionally links the mRNA of NLMs to brain physiopathology. Furthermore, the results demonstrate that the RNAs associated with NLM contain lincRNA and snRNA genes. The first influence remodeling chromatin and genome architecture, RNA stabilization, and transcription regulation, including enhancer-associated activity[48]. The second are critical components of the spliceosome that participate in pre-mRNA splicing[49].

Another central point raised by this study concerns a higher percentage concentration of retained introns with respect to that of total nuclear RNAs. It has been demonstrated that the retained introns associated to the lamin are rich in genes that work in RNA processing, translation, and the cell cycle[50]. Here, the retained introns by including exons allow the formation of EIciRNAs that regulate the transcription process[23]. It has been demonstrated that NLMs are located in the inner nuclear membrane associated with the lamin[9] and are rich in RNase-resistant RNA that becomes sensitive to the enzyme after thermal denaturation or degradation of SM with nSMase treatment, indicating its non-linear structure[14].

Interestingly, retained introns can increase the association of putative miRNA targets in mRNA[51]. Although mature miRNAs are primarily known for their cytoplasmic role in gene silencing, they can also undergo nuclear retrotransport, where they regulate transcription and RNA processing[52]. This process involves multiple transport mechanisms, including Importin-α/β, Exportin-1 (XPO1/CRM1), and RNA-induced silencing complex

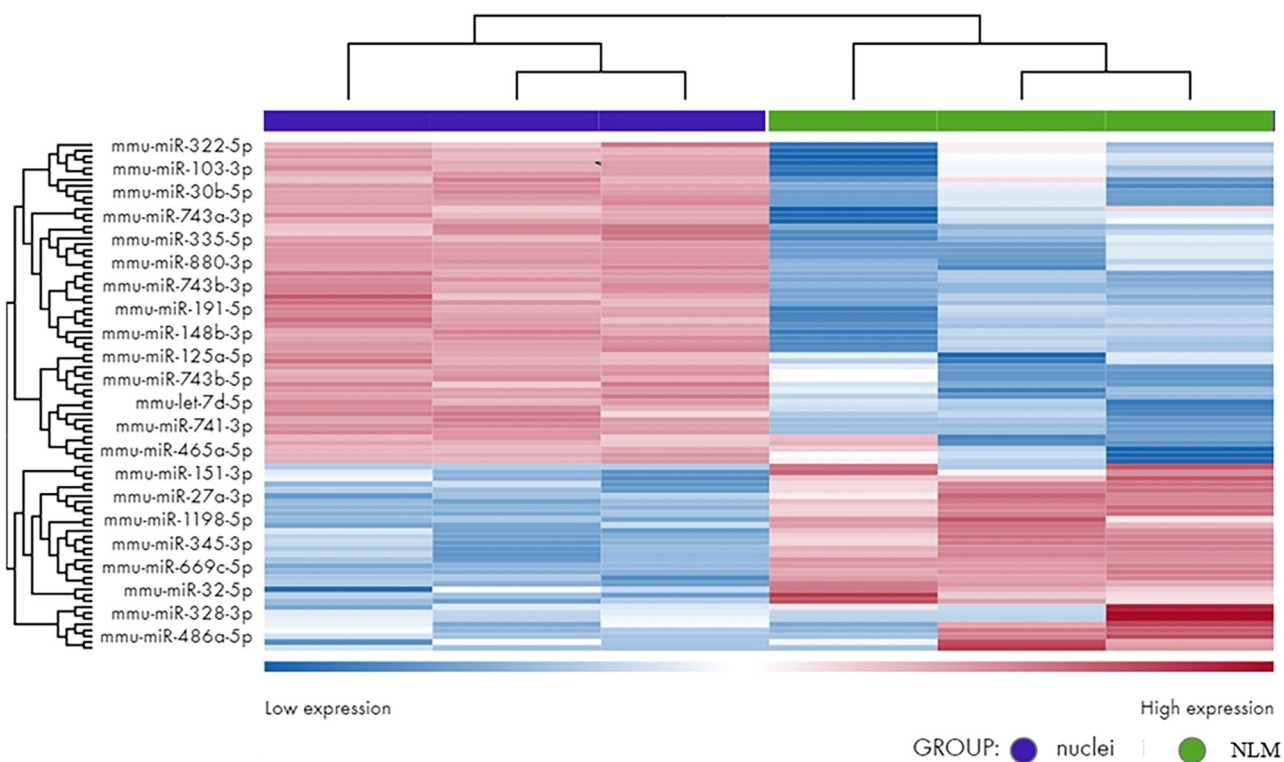

**Fig. 9 | miRNA profiling of nuclear lipid microdomains (NLM) compared to nuclei.** Heatmap showing the statistically significantly differentially expressed miRNAs in NLMs respect to the nuclei. Globally, twenty-two miRNA were differentially present, $n = 14$ low-concentrated and $n = 8$ high-concentrated in NLM in comparison with nuclei (abs(logFC) ≥ 0.6, $q$-value ≤ 0.05). Hierarchical clustering of transcripts and samples using the Euclidean distance and the complete agglomeration method; expression data was VST-transformed, scale,d and centered. When not available, the miRBase accession number was replaced by the ENSEMBL GeneID. The heatmap was generated using the DESeq2 R package (v.1.0.12).

(RISC) components like Argonaute 2 (AGO2)[44,45,47–53]. Once in the nucleus, miRNAs influence key processes such as transcriptional regulation, alternative splicing, and interactions with nuclear RNAs. For instance, miR-9 and miR-122 modulate gene expression by interacting with promoters and RNA polymerase II, respectively[27,54].

Additionally, exosomes play a crucial role in intracellular miRNA trafficking, potentially storing and releasing miRNAs into the cytoplasm or nucleus in response to specific signals. They may facilitate nuclear miRNA localization by interacting with nuclear lipid microdomains (NLMs) and transport proteins[54]. This highlights the dynamic role of miRNAs beyond cytoplasmic gene silencing, contributing to transcriptional and epigenetic regulation. While this field is still evolving, growing evidence suggests that nuclear miRNA trafficking is a key mechanism in cellular homeostasis and disease.

Our results showed that NLMs are rich in miRNAs, so much so that they have a similar value to mRNA. We demonstrated that the concentration of miRNA is very high in NLMs, particularly 151-3p, 27a-3p, 1198-5p, 345-3p, 669c-5p, 32-5p, 328-3p, 486a-5p. It has been described that miRNA 151-3p regulates synaptic reorganization by influencing long-term memory processes as input specificity, rapid induction, cooperativity, and associative interactions[55]. miRNA 27a-3p is involved in the retinoblastoma progression[56], in the promotion of cerebral ischemia/reperfusion injury[57], in the modification of hypoxia-induced malignant behaviors of glioma cells[58], in the neurotoxicity induced by anesthetic administration[59], in the multidrug resistance P-glycoprotein in blood-brain barrier[60] and in the promotion of neurofibromatosis[61]. miRNA 345-3p is recognized to play a role in the glioblastoma progression[62]. miRNA 669c-5p plays a role in the traumatic brain injury[63]. miRNA 32-5p mediates neuropathic pain and neuroinflammation[64]. miRna-328-3p is correlated with mesial temporal lobe epilepsy with hippocampal sclerosis[65]. miRNA 486a-5p has an impact on microglial cell viability[66]. The nuclear localization of miR-328, consistent with its retrotransport mechanisms, supports its role in transcriptional regulation and RNA processing[67]. This finding aligns with its involvement in multiple sclerosis, where it influences oligodendrocyte differentiation and myelination, as well as in neurodegenerative diseases, where it regulates neuroinflammation and oxidative stress. Additionally, in glioblastoma, its nuclear presence may contribute to tumor suppression by modulating gene expression and cellular invasiveness.

The high concentration of miRNA in NLMs may be justified by the simultaneous presence of EIciRNAs, which, with their retained introns, serve as sponges for miRNA. EIciRNAs of NLMs contain ZFX, STRN3, TRUB1, CCNT2, PKP2, CTNNA1, CYREN, DEPDC5, SLC24A2, CSTF3, ACACA, ASCC1, KMT2D, ERCC6, PPP1R9A, PPFIBP1, ANP32P, CCDC171 genes. The ZFX gene is associated with neuroblastoma[25] and X-linked neurodevelopmental disorder[68]. STRN3 is related to a de novo variant in *PPP2CA* in a patient with a developmental disorder, autism, and epilepsy[69]. TRUB1 and PKP2 are involved in the development of glioblastoma[70,71]. CCNT2 promotes oligodendrocyte differentiation[72]. CTNNA1, KMT2D and PPFIBP1 are involved in schizophrenia[73–75], DEPDC5 in epilesy[76], SLC24A2 and ACACA in neurodegenerative disorders[77,78], ASCC1 in neuromuscular syndrome[79] and ERCC6 in cockayne syndrome[80]. Additionally, NLMs host 3 ciRNA, notably regulating the transcription of their parental genes[19,20], exactly RECQL5, IRAK1, and KMT2D genes. RECQL5 and IRAK1 are correlated with depressive behavior[46,81] and KMT2D with schizophrenia[71], as reported for EIciRNA.

Taken together, the data obtained clearly indicate that all types of RNAs resistant to RNase, which remain associated with NLMs, are involved in chromatin organization and brain pathophysiology. Nuclear SM plays a crucial role in maintaining the double-stranded and/or circular structure of RNA, as treatment with SMase, leading to SM degradation and consequently Chol detachment, results in the loss of RNase resistance.

Thus, this functional interplay between RNAs associated with NLMs of embryonic hippocampal cells and brain physiopathology, introduces a novel dimension to the regulation of nuclear activity.

Our results demonstrated that incubation of cells with SM and Chol in equimolecular concentration resulted in an enrichment of saturated SM and Chol. It is known that saturated SM is able to bind Chol with Van der Waals forces in order to form lipid microdomains or rafts. Functionally, the enrichment of SM-Chol NLMs was associated with an increase in RNase-resistant RNA content and cellular differentiation. It is very difficult at the moment to establish the cause-and-effect relationship between the enrichment of saturated SM and Chol as well as RNA resistant to the action of RNase in NLMs, and the differentiating action of the incubation of cells with SM-Chol. Considering the data obtained from the functional analysis of the RNAs present in the NLMs towards neuronal differentiation, it is possible to hypothesize that the SM and Chol that entered the cell and managed to reach the NLMs were able to act as a basis for the control of the regulation of genes involved in cell differentiation. As a consequence, the cells differentiated. On the other hand, our previous results demonstrated that NLMs served as a platform for anchoring active chromatin. Subsequent studies on molecular chemical-physical interactions will be able to clarify the exact mechanism of accumulation of RNAs in NLMs. In fact, at the moment, it is really difficult to establish how mRNAs and miRNAs are getting associated with NLMs. It is known that polar lipid-DNA interactions are due to the electrostatic interactions between DNA and membrane headgroups. Interestingly, the affinity of DNA for membrane phospholipid groups depends on whether it is single-stranded (ssDNA) or double-stranded (dsDNA). Specifically, dsDNA has a higher affinity than ssDNA[82]. Moreover, the association of tRNA or DNA with polar lipids is linked to cationic-anionic group interactions[83,84]. Nothing is known about the polar lipids of nuclear RNA. It is possible to hypothesize that there is an interaction between the polar head of the SM and the hydroxyl groups of the ribose of the RNA.

In summary, based on these data, this represents a hitherto unrecognized role for nuclear SM, underscoring a close association between NLMs and RNase-resistant RNA, such as dsRNA, EIciRNA functioning as a miRNA sponge, and ciRNA. These findings have implications for advancing our understanding of the diverse mechanisms governing gene expression.

It is conceivable that the activation of nuclear SMase in response to cellular signals may induce a conformational modification of dsRNA and/or cirRNA, resulting in the activation of the transcriptional machinery[82]. This activation could lead the cell either towards physiological proliferation/cancer/degeneration or towards differentiation/apoptosis. This mechanism could elucidate the role of nuclear SM in neurodevelopment[85], neurodegeneration[86], apoptosis[6,87,88], cancer[31,89]. Moreover, numerous studies highlight the role of SM and/or its metabolism in various pathologies, elucidating modifications in specific genes. The majority of these investigations have been conducted on whole cells, assessing total SM levels, including the nuclear compartment. In light of our findings, it is conceivable that nuclear SM is specifically involved in the alteration of gene expression regulation underlying different pathologies.

## Materials And Methods

### Experimental plan
The experimental plan is illustrated in the following diagram (Fig. 10).

### Reagents
Dulbecco's modified Eagle's medium (DMEM), penicillin, streptomycin L-glutamine, trypsin-ethylenediaminetetraacetic acid disodium and tetrasodium salt (EDTA) solution, fetal bovine serum (FBS) were from Microgem srl (Pozzuoli, NA, Italy); bovine serum albumin was from Thermo Fisher Scientific (Waltham, MA, USA); SM, Chol, nSMase, PC-PLC, acetonitrile, 2-propanol, methanol, ethanol, chloroform, formic acid, ammonium acetate, and ammonium formate were from Sigma-Aldrich (St. Louis, MO, USA). 16:0 SM, 18:1 SM, 24:0 SM, were purchased from Avanti (Avanti Polar, Alabaster, AL); anti-giantin, anti-STAT3 were obtained from Santa Cruz Biotechnology (Santa Cruz, CA); anti-lamin B was obtained from Oncogene (Boston, MA). Purified water was from Milli-Q grade (Burlington, MA, USA). For research involving biohazards, biological select agents and reagents, standard biosecurity safety procedures were carried out.

### Cell culture and treatments
Immortalized hippocampal neurons HN9.10e (kind gift of Kieran Breen, Ninewells Hospital, Dundee, UK) were grown in DMEM supplemented with 10% FBS, 2 mM L-glutamine, 100 IU/mL penicillin, 100 μg/mL streptomycin, and 2.5 μg/mL amphotericin B[90]. To test the effect of SM and Chol on NLM composition, $500 \times 10^3$ cells were incubated with 5, 10, 15,

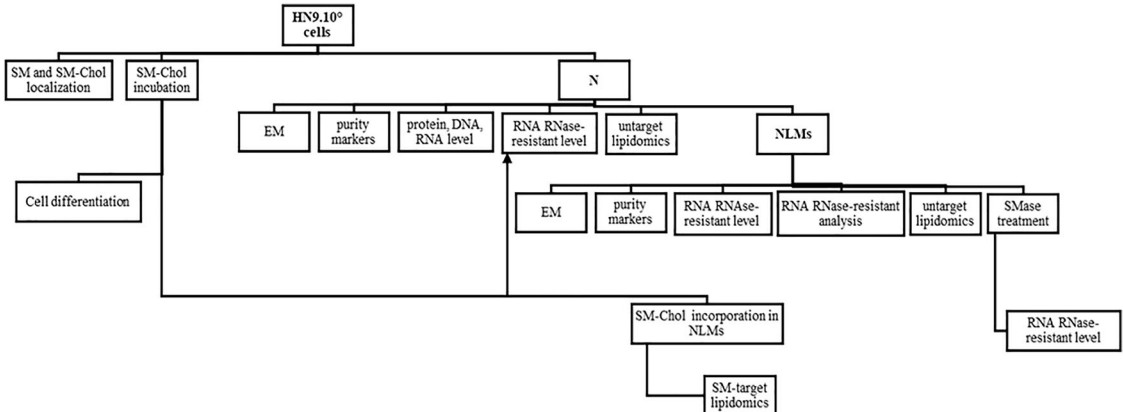

**Fig. 10 | Embryonic hippocampal cells (HN9.10e) were used as an experimental model.** Cells: (1) were used to locate Sphingomyelin (SM) and SM-cholesterol (Chol) microdomains by EGFP-NT-Lys and EGFP-NAK fluorescent probes, respectively; (2) were incubated with increasing concentration of SM and Chol to study their role in cell differentiation and in RNase-resistant RNA content increase in nuclear lipid microdomains (NLMs). The achievement of NLMs by cell-incorporated SM-Chol was monitored by total SM and Chol assays and by target lipidomics for SM; (3) were used to purify nuclei. Nuclei were used: 1) for electron microscopy (EM) analysis; 2) for the control of their purification level by analyzing Giantin as a Golgi marker; 3) for the analysis of their composition in proteins, DNA and RNA characteristic of purified nuclei; (4) for the analysis of all types of RNA; 4) for the analysis of lipid composition by untargeted lipidomics; 5) for the purification of NLMs. NLMs were used: 1) for electron microscopy (EM) analysis; 2) for the control of their purification level by analyzing lamin b and STAT3 as their specific markers; 3) for the analysis of total RNase-resistant RNA content; 4) for the analysis of RNA RNase-resistant composition; 5) for the analysis of lipid composition by untargeted lipidomic; 6) for the evaluation of the effect of SMase on the RNase-resistant RNA content.

20 μM SM-Chol prepared in equimolar ratio as liposomes for 6 h. The NLMs were purified and analyzed.

## Cell morphology

HN9.10e cells were cultured for 72 h in the absence or presence of 20 μM SM-Chol. Then, they were stained with methylene blue. The observation was performed by using inverted microscopy EUROMEX FE 2935 (ED Amhem, Netherlands) equipped with a CMEX 5000 camera system (40x magnification) and the morphometric analysis was performed by using ImageFocus software (EUROMEX, Arnhem, The Netherlands).

## Electron microscopy

Electron microscopy analysis was performed as previously reported[15]. Pellets of nuclei, nuclei treated with Triton X-100 and NLMs, prepared as reported below were fixed in 2.5% glutaraldehyde in 0.1 M sodium phosphate buffer, pH 7.2, for 2 h at 4°C. After washing in 0.1 M sodium phosphate buffer, the samples were treated with osmium tetroxide in 0.1 M sodium phosphate buffer for 1 h at 4°C, dehydrated in graded concentrations of acetone, and finally embedded in Epon. Ultrathin sections were cut on a Reichert-Jung Ultracut E (ultramicrotome), stained in 1% uranyl acetate in water and lead citrate, and finally examined in a Philips EM208 electron microscope (Electronic Instruments, Mahwah, NJ) equipped with a camera system at a constant temperature of 18 °C and 60 KW high tension.

## Fluorescence

The SM or SM/Chol complex localization was studied by using the specific probes.

For SM, enhanced green fluorescent protein-nontoxic-lysenin (EGFP-NT-Lys) was used[91,92]. The probe was purified from Ecoli strain BL21(DE3) harboring pET28/EGFP-NT-Lys. In brief, after the bacteria culture reaches OD600600 = approx. 1, the expression of EGFP-NT-Lys was induced at 18 °C for two overnights in the presence of 125 μM IPTG (VWR Life Science). EGFP-NT-Lys was purified by Hitrap TALON crude column (Cytiva, Munzinger Str. 5, 79111 Freiburg im Breisgau, Germania) from bacterial lysate using its His-tag. Imidazole, used in the elution of the protein from the column, was removed by dialysis. The dialyzed protein was mixed with glycerol (VWR chemicals) at 50% (v/v) and stored at –20 °C. On the day of the experiment, the medium was removed, cells were washed with DMEM/5% lipid-depleted serum (LPDS) and treated with 15 μg/mL EGFP-NT-Lys diluted in DMEM/5% LPDS for 2 h, Then, cells were fixed with 250 μL of 4% paraformaldehyde (PFA) at room temperature (RT) for 30 min. After washing, the residual PFA was neutralized by 0.1 M NH4Cl at RT for 15 min. Cells were washed three times with 500 μL of PBS and nuclei counterstained with DAPI. Coverslips were mounted and cells viewed in a DMRB Leica epimicroscope equipped with a digital camera.

For SM/Chol complex detection, green fluorescent protein-nakanori (EGFP-NAK) probe was used. The probe was purified from *E.coli* strain BL21(DE3) harboring pET28/EGFP-NAK according to Panevska et al.[93] with little modifications. In brief, 0.5 ml of overnight preculture of the above bacteria in 2 ml of LB broth (Sigma-Aldrich) supplemented with 50 μg/ml kanamycin (LB + kan) was inoculated in 500 ml LB + kan. After the bacteria culture reaches OD$_{600}$ = 0.6–0.8, the expression of NAK was induced at 20°C overnight in the presence of 0.5 mM IPTG (VWR Life Science). NAK was purified by Hitrap TALON crude column (Cytiva) from bacterial lysate using a His-tag at the N-terminus. Imidazole used in the elution of the protein from the column, was removed by dialysis. The dialyzed protein was mixed with glycerol (VWR chemicals) at 50% (v/v) and stored at −20 °C.

For cell differentiation evaluation, cells were incubated with anti-GFAP or anti-βtubulin III primary antibodies diluted 1:100 in 3% (w/v) BSA in PBS for 1 h, washed three times in 0.1% (v/v) Tween-20 in PBS and twice in PBS, incubated with tetramethylrhodamine isothiocyanate (TRITC)-conjugated anti-rabbit IgG for 1 h, diluted 1:50 in 3% (w/v) BSA in PBS and washed as above. The 4′,6-diamidino-2-phenylindole (DAPI) nuclear counterstain was used. The samples were mounted in 80% (w/v) glycerol, containing

0.02% (w/v) NaN$_3$ and p-phenylenediamine (1 mg/mL) in PBS to prevent fluorescence fading. The antibody incubations were done in a humid chamber at room temperature.

Fluorescent analysis was performed on a DMRB Leika epi-fluorescent microscope equipped with a digital camera. The intensity of immunofluorescence was evaluated with Scion Image.

## Homogenate preparation of cell culture

Cells were washed two times with Phosphate Buffered Saline (PBS) and centrifuged at 1000 g for 10 min. The pellets were suspended in buffer containing 3 mM Tris–HCl, 3 mM CaCl$_2$, 2 mM Mg acetate, 0.5 mM dithiothreitol + 1 mM phenylmethylsulfonyl fluoride (PMSF), pH 8.0 (2.5 ml/10 x 10^6 cells) and gently homogenized by a tight-fitting teflon-glass homogenizer.

## Nuclei isolation from cell culture

The nuclei were isolated according to Bartoccini et al.[18] with modifications. Triton X-100 (1% in the above buffer) was added to homogenate (0.5:1 v/v). Sample was stirred on a vortex mixer for 30 s and the 1.5 M sucrose in the above buffer was added (0.25:1 v/v). After centrifugation at 2000 g × 10 min, the pellet containing the nuclei was washed twice with 0.1 M Barnes solution containing 0.085 M KCl, 0.0085 NaCl, 0.0025 M MgCl$_2$, trichloroacetic acid-HCl 0.005 M. The purity of nuclei was controlled by evaluating the glucose-6-phosphatase activity and NADH-cytochrome C-reductase activity, as previously reported[18]. The final pellet was resuspended in 0.1 M Tris, pH 7.2, and used in part for biochemical analyses and in part for nuclear lipid/RNA-rich microdomain purification.

## Nuclear lipid microdomain extraction

NM was prepared according to Rossi et al.[16] HN9.10 cell nuclei were washed in 10 mM Tris, containing 0.25 M sucrose, 2.5 mM MgCl$_2$, 0.5 mM PMSF, and 1% Triton X-100, pH 7.5, at 0 °C, in order to remove the external nuclear membrane, preserving the inner nuclear membrane, and were centrifuged at 1000 g for 15 min at 4 °C. The pellet was washed to times with 0.1 M Barnes solution at pH 7.4 and re-suspended in the same solution. Then the nuclei thus treated were digested with DNase I (120 μg/ml) and RNase cocktail (500 μg/ml) for 15 min at 37 °C. The reaction was stopped on ice with Barnes solution (1:1 by vol.). After centrifugation at 5000 rpm for 5 min, the pellet was resuspended in 30 mM Tris containing 0.4 M ammonium sulfate, which facilitated the RNA precipitation, and 1 mM PMSF, pH 8.4, and then was centrifuged at 10,000 rpm for 15 min.

## Sphingomyelinase and phosphatidylcholine-dependent phospholipase C treatment

For SMase treatment, samples of NM corresponding to 25 μg of RNA were incubated with 0.1 M Tris–HCl, pH 8.4, 6 mM MgCl$_2$, 0.1% Triton X-100 in the presence of 5 μl (6 U/mg prot) of N-SMase at 37 °C to a final volume of 0.5 ml. For PC-PLC treatment, samples of NM corresponding to 25 μg of RNA were incubated with 0.1 M Tris–HCl, pH 8.4, 2 mM CaCl$_2$, 0.1% Triton X-100 in the presence of 10 μl (6 U/mg prot) of PC-PLC at 37 °C to a final volume of 0.5 ml. After 60 min RNase-cocktail (500 μg/ml) was added and the reaction was carried out for a further 30 min. The reaction was stopped on ice, and the amount of RNA, precipitated with absolute ethanol, was evaluated and compared with that of the sample incubated without SMase or PC-PLC.

## Biochemical analyses

Total protein, DNA, and RNA amounts in whole nuclei and in NLsM were analyzed[18].

## Western blotting

Thirty micrograms of protein from NLMs were submitted to SDS–PAGE electrophoresis in 8% polyacrylamide slab gel for giantin detection and 10% gel for STAT3 and lamin B according to Lazzarini et al.[94] The transfer of protein was carried out onto nitrocellulose in 90 min. The membranes were

blocked for 30 min with 5% nonfat dry milk in PBS (pH 7.5) and incubated overnight at 4 °C with specific antibodies (anti-giantin and anti-STAT3 diluted 1:1000, anti-lamin B diluted 1:2000). The blots were treated with horseradish-conjugated secondary antibodies for 60 min. Band detection was performed using an enhanced chemiluminescence kit from Amersham Pharmacia Biotech (Rainham, Essex, UK). A densitometric analysis was performed by Chemidoc Imagequant LAS500–Ge Healthcare-Life Science (Milano, Italy).

## Untargeted lipidomics

Lipids from nuclei and NLM preparation (less than 100 μg of proteins) were extracted by a water/methanol/chloroform mixture (30:60:10, v/v/v). LC-MS/MS consisted of an Agilent 1290 Infinity II LC (Agilent) connected to a ZenoTOF 7600 System (SCIEX) equipped with a Turbo V™ Ion Source with ESI Probe. All samples were analyzed in duplicate in both positive and negative mode with electrospray ionization. Spectra were contemporarily acquired by full-mass scan from 200–1500 m/z and top-20 data-dependent acquisition from 50–1500 m/z. Declustering potential was fixed to 50 eV, and the collision energy was 35 ± 15 eV. The chromatographic separation was reached on a reverse-phase Acquity CSH C18 column 1.7 μm, 2.1 × 100 mm (Waters, Franklin, MA, USA) equipped with a precolumn by a gradient between (A) water/acetonitrile (60:40) and (B) 2-propanol/acetonitrile (90:10), both containing 10-mM ammonium acetate and 0.1% of formic acid Details can be found in Dei Cas, M. et al.[95]

## LC-HR-MS data processing

The spectra deconvolution, background subtraction, peak alignment, and sample normalization were attained using MS-DIAL (ver. 4.0). MS and MS/MS tolerance for peak profile were set to 0.01 and 0.05 Da, respectively. Identification was achieved by matching spectra with the LipidBlast database. Intensities of analytes were normalized by Lowless algorithm, and those with a CV% superior to 30% in the QC pool sample were excluded.

## Lipid analysis in nuclear lipid microdomains purified from HN9.10e cells treated with increased concentration of SM-Chol

The total SM content was analyzed in NLMs purified from HN9.10e cells treated with increased concentration of SM-Chol, as above reported. From each sample, total lipids were extracted with chloroform/methanol (2:1, v/v), as previously reported[15]. The phospholipids (PLs) were separated by thin-layer chromatography (TLC) on silica gel using chloroform:methanol: ammonia, 65:25:4 vol/vol/vol. The SM was detected with iodine vapors using commercial SM as a standard. The spot was scraped into test tubes for inorganic phosphorus determination[15]. Sphingolipidomic analysis was performed in NLM purified from HN9.10e cells treated with 20 μM SM-Chol by using the Ultra Performance Liquid Chromatography system tandem mass spectrometer (Applied Biosystems, Italy), as previously reported[95]. 12:0 SM was used as an internal standard. The 12:0 SM, 16:0 SM, 18:1 SM, and 24:0 SM standards were dissolved in chloroform/methanol (9:1 vol/vol) at 10 μg/ml final concentration. The stock solutions were stored at −20°C. Working calibrators were prepared by diluting stock solutions with methanol to 500:0, 250:0, 100:0, and 50:0 ng/ml final concentrations. Twenty microliters of standards or lipids extracted from HN9.10e cells was injected after purification with specific nylon filters (0.2 μm). The samples were separated on a Phenomenex Kinetex phenyl-hexyl 100 A column (50 × 4.60-mm diameter, 2.6-μm particle diameter) with a precolumn security guard Phenomenex ULTRA phenyl-hexyl 4.6. Column temperature was set at 50°C and flow rate at 0.9 ml/min. Solvent A was 1% formic acid; solvent B was 100% isopropanol containing 0.1% formic acid. The run was performed for 3 min in 50% solvent B and then in a gradient to reach 100% solvent B in 5 min. The system needed to be reconditioned for 5 min with 50% solvent B before the next injection. The SM species were identified by using positive turbo-ion spray and modality multipole-reaction monitoring. The identification and analysis of Chol was conducted by atmospheric pressure chemical ionization in positive ionization conditions and multipole-ion scan modality[95].

## RNA extraction and library preparation and sequencing

Total RNA was extracted from 3 nuclear and 3 NLM samples using the miRNeasy Tissue/Cells Advanced Micro kit, according to the manufacturer's instructions. RNA was recovered with 30 μL of nuclease-free water. RNA samples were quantified by using the Qubit 2.0 Fluorometer (Invitrogen) using the high-sensitivity RNA (HS) assay (Invitrogen). A total amount of 50 ng was used for library preparation using the "QIAseq® FastSelect™ RNA Library Kit" (Qiagen), following the manufacturer's instructions.

Briefly, total RNA was thermally treated to allow QIAseq FastSelect to remove ribosomal RNA. Then, the treated RNA was reverse-transcribed into cDNA using both random hexamers and oligo-dT primers for a complete RNA coverage. Unique sample indexes were incorporated to each sample transcript during the same transcription process. Following cDNA synthesis, each library was amplified, and unique dual indices were added using the QIAseq UX Index Kit (Qiagen).

Libraries were quantified by the Qubit 2.0 Fluorometer (Invitrogen) using the high-sensitivity DNA assay kit, and a Bioanalyzer 2000 (Agilent) high-sensitivity assay was used to check for the expected size distribution of library fragments. Single libraries were pooled and then sequenced on paired-end 74 bp mode on an Illumina NextSeq instrument (Illumina).

## Validation of sequencing results by RT-qPCR

Total RNA extracted from three nuclei and four NLMs, as described in the previous section, was reverse transcribed using the Superscript III enzyme (Applied Biosystems, Thermo Fisher Scientific) following the manufacturer's instructions. The resulting cDNA was analyzed by RT-qPCR using the LightCycler 480 system (Roche) to confirm differential mRNA and circRNA expression. Specific primers were designed for 3 target genes (Ehmt2, Chaf1a and Dnm2) and 4 target circRNAs (Slc24A2, Ccnt2, and Deodc5) (Supplementary Table 3). Two murine housekeeping genes (Gapdh and Actin beta) were selected for normalization of the Cycle threshold (Ct) values. Fold changes were calculated using the ΔΔCt method.

For miRNA validation, qRT-PCR was performed using the TaqMan Advanced miRNA Assay for mmu-miR-27a-3p, miR-328-3p, and mmu-miR103-3p (Applied Biosystems, Thermo Fisher Scientific), with 10 μL of total RNA per reaction in a LightCycler 480 system (Roche). All reactions were run in triplicate. As there is no current consensus on the selection of housekeeping microRNAs for qRT-PCR analysis, no normalization was applied.

## Bioinformatics analysis

Three biological replicates were analyzed for each condition, averaging 15 million sequenced reads per sample. For RNA-sequencing data analysis, FASTQ files were uploaded to QIAGEN GeneGlobe for use with the RNA-seq Analysis Portal. Here, raw data were processed for format conversion, sample demultiplexing, differential gene expression, and pathway analysis. Upon adapter removal, fragments were mapped to the reference genome, and differentially expressed-RNAs were discovered using the DESeq2 algorithm. An FDR value < 0.01 data significance threshold was applied to allow the identification of differential RNAs expression.

## RNAs functional enrichment analysis

QIAGEN Ingenuity Pathway Analysis (IPA) was used to characterize the functions of differentially expressed RNAs. Differential RNA expression was calculated and visualized using volcano plots. Differentially expressed RNAs were queried against the QIAGEN Knowledge Base for canonical pathways, upstream regulators, diseases, and biological functions. Z-scores ‹-2 and ›+2 were assumed to identify the most significantly enriched terms.

Mouse validated target genes of upregulated miRNAs, between NM and nuclei, were retrieved from Tarbase [https://pubmed.ncbi.nlm.nih.gov/29156006/]. Upregulated miRNAs target genes were then filtered to maintain only genes that were significantly upregulated between NM and nuclei. Enrichment analysis was then performed using the Gprofiler2 R package with default parameters by querying the Gene Ontology database

[https://pubmed.ncbi.nlm.nih.gov/33564394/, https://pubmed.ncbi.nlm.nih.gov/14681407/]. Significant results were identified as ontologies having an adjusted *p-value < 0.05*.

## Circular RNAs detection from RNA-seq experiment

For circular RNAs annotation, we take advantage of the Circexplorer2 software (https://genome.cshlp.org/content/26/9/1277.abstract) with default parameters, specifically using the STAR software for read mapping (https://pubmed.ncbi.nlm.nih.gov/23104886/). Briefly, reads were mapped to the reference *Mus musculus* genome (version GRCm38 downloaded from Gencode) using STAR by allowing the detection of chimeric (fusion) alignments (--chimSegmentMin 10, allowing a minimum mapped length of 10). Then, mapped reads were parsed for the presence of chimeric reads using the command "parse" of the Circexplorer2 software, and finally, circular RNAs were annotated using the "annotate" Circexplorer2 command (https://genome.cshlp.org/content/26/9/1277.abstract).

## Statistics and reproducibility

Data were expressed as means ± SD of different numbers of samples in relation to the experiments, as reported in the figure legends. Significance was verified by Student's *t*-test (experimental samples versus control sample). For RNA and LC-HR-MS, statistical analysis was above reported.

## Reporting summary

Further information on research design is available in the Nature Portfolio Reporting Summary linked to this article.

## Data availability

The source data behind the graphs in the paper can be found in "Supplementary Data". The datasets generated and/or analyzed during the current study are available in the NCBI GEO repository, link https://www.ncbi.nlm.nih.gov/geo/query/acc.cgi?acc=GSE262449.

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

## Acknowledgements

Part of this work was carried out in OMICs, an advanced mass spectrometry platform established by the Università degli Studi di Milano. We thank Giulia Corso for the preparation of the Graphic Abstract.

## Author contributions

The conception and design of the study F.C. and E.A.; experiments C.C., M.B., F.F., S.C., N.G., M.G.G., G.V., R.P., M.D.C., C.A., A.M., T.K., N.T.; acquisition of data C.C., M.B., F.F.; analysis of data O.C., M.B., T.B., E.A.; interpretation of data F.C. and E.A.; drafting the article or revising it critically for important intellectual content E.A., F.C., M.B., P.S.; final approval of the version to be submitted E.A., F.C. All authors have read and agreed to the published version of the manuscript.

## Competing interests

The authors declare no competing interests.
