## [Transparent Peer Review file · Communications Biology]

Sphingomyelin regulates the transcriptional machinery in nuclear lipid microdomains

Corresponding Author: Professor Elisabetta Albi

Version 0:

Reviewer comments:

Reviewer #1

(Remarks to the Author)

This manuscript describes the importance of sphingomyelin and cholesterol in preserving nuclear RNA. The authors profiled nuclear lipid microdomain RNA, and identified various mRNAs, lncRNAs, microRNAs and circular RNAs. They also found that sphingomyelinase treatment enhanced the effect of RNase treatment. All of the authors' conclusions are correlative and do not provide any strong conclusion. In order to develop this study further, the authors should provide experimental data supporting the consequence of RNA localization in the microdomains. Enhancement or inhibition of those RNA accumulation in the microdomains and the subsequent outcomes in cellular process (cell division cycle, proliferation, death, etc.) should be tested before publication. Molecular mechanisms of how those RNAs accumulate in the microdomain should be characterized even further.

Reviewer #2

(Remarks to the Author)

In this manuscript entitled "Sphingomyelin and RNase-resistant RNA in Microdomains of the Inner Nuclear Membrane", Conte & Bulfoni et al., characterize the RNAs enriched in nuclear microdomains (NLMs) that are resistant to RNase treatment. Using the neuronal cell line HN9.10, the authors found that sphingomyelin and cholesterol protect certain mRNAs, miRNAs, and circRNAs against degradation. As a result, these RNAs became enriched in these nuclear sub-compartments and are believed to have gene expression regulatory functions important for neuronal health. This paper's findings are interesting, but it has several issues, related to experimental controls, data missing, and description of experimental design and results that need to be solved before publication.

Besides this general comment, the paper will benefit from addressing the following concerns

1. The rationale beyond using the embryonic hippocampal cells needs to be explained because previous studies have been done in hepatocytes. It should be explained if the finding of RNAs in NLMs related to neuronal pathologies is a direct consequence of the cell line used in these studies.
2. The statistical test, n, and significance should be provided for all figures.
3. Figure 2 misses sufficient experimental details and quantification to support the conclusion. Only 2 cells are shown per analysis, and they look different from each other. There is no quantitative data for the immunofluorescences or indication of the number of cells or experiments done. The figure also misses a negative control and a marker for the cell plasma membrane. The scale bar should be added to the figure.
4. Figure 3, how does incubation with SM-Chol result in their increased content in the nucleus? This is not quantified or demonstrated, and it is needed to support the conclusions from these sections.
5. Figure 4, in the text lines 233-235, Why the percentages of RNA classes do not sum to 100% and are different from the figure? What statistics were applied to support that there are differences between the nuclear and NML RNAs?
6. Relate with the conclusion from Figure 4 in which the authors found that mRNAs are less enriched in NLMs than in the nuclei. What is the rationale for analyzing mRNAs? And Why the number of enriched mRNAs in NLMs is similar to the number of microRNAs?
7. Figures 5 and 7, the findings by RNAseq should be validated by RT-qPCR. The meaning of the pools should be explained in the figure legend.
8. Figure 6, how many mRNAs are per Go-term category?

9. In the sentence "The functional analysis of genes highly concentrated in the NLMs compared to the nuclei, which are 281 targeted by miRNAs at high concentrations in the NLMs compared to the nuclei....." The meaning of targeted by miRNAs is not well explained. In the legend, there is a typo in "intensity"
10. This sentence needs references, probably a dot. "The association between these 307 miRNA and diseases was reported in Figure S5. It is evident that they are correlated with neuro psychological diseases as chronic epilepsy, amyotrophic lateral sclerosis and psychological disorders "
11. Data in Figure 8 need validation by RT-QPCR and demonstration that the RNAs are in fact circular RNAs.
12. Discussion line 348. It will be helpful to have an explanation as for why the authors focused on mRNAs when circRNAs and miRNAs were previously found.
13. There is inconsistent formatting in the materials and methods section.

Reviewer #3

(Remarks to the Author)

In their manuscript Conte et. al., Has come up with an idea of nuclear lipid microdomain with the inner nuclear membrane to host different classes of RNA - protected from RNases due to sphingomyelin. The biochemical isolation has been done to separate NLM-associated pool from total cellular pool. The authors have found candidate RNAs which are enriched for NLMs. This data have potential implications in physiological processes such as chromatin organization and brain disease pathology.

Although the hypothesis is interesting and potentially exciting to consider NLM as important sites for storage of miRNA and other RNAs for regulation of gene expression, the work is incomplete and misses several important controls that dampen my enthusiasm for this manuscript.

There are several issues that need attention:

The isolation of Nuclei and Nuclear associated microdomain should be confirmed by Western blot analysis for suitable marker proteins. This is important to conclude on the purity of the fractions they have studied in subsequent analysis and RNA estimation.

It is essential also to perform super resolution microscopic analysis and TEM to comment on the size and shape of the microdomain with RNA within cells and after isolation as NLM. The microscopic data provided is not convincing for the reviewer.

miRNA s are synthesized as pri- miRNAs and processed inside the nucleus to pre-miRNAs which are subsequently get exported to cytoplasm to get processed by Dicer1 in human cells cytoplasm to mature miRNA. I wonder what form of miRNAs (mature or pre-miRNA) that the authors detected with NLM. If I understand correctly, the mature miRNAs are followed. This is unusual as only a small fraction of miRNA is detected nucleus in normal condition. Therefore, if it is mature miRNA , then how they were getting retro-transported to nucleus from cytoplasm.

Schematic pictures depicting the research methodology and experimental flow should be included for better understanding the separation processes.

Version 1:

Reviewer comments:

Reviewer #1

(Remarks to the Author)

The authors examined the morphological changes and differentiation phenotypes of HN9.10e cells. However, they still did not examine molecular mechanism of how exactly these events happen. Thus, this manuscript is still under the process of further development.

Reviewer #2

(Remarks to the Author)

The authors have addressed the reviewers' comments; however, the quality of the experimental data should be improved. For example, the western blots in Figure 1 and the immunofluorescence images in Figure 4 appear overexposed, and the latter lacks a scale bar. It is also unclear whether the cells in the images are viable.

Reviewer #3

(Remarks to the Author)

The authors have partly addressed my concerns in the revised manuscript. However, the manuscript still lacks any mechanistic information on the compartmentalization of RNAs to lipid-rich membrane microdomains of the nuclear membrane. I wish the authors would provide a hypothesis, at least in the discussion part, on how these mRNAs and miRNAs are getting associated with lipid-rich domains. These findings may have physiological importance and could thus be accepted for publication, but with additional textual changes introduced.

Dear Editor,

Thank you very much for your letter and referees' comments. Accordingly, we have revised the manuscript.

Reviewers' comments:

Reviewer #1 (Remarks to the Author):

This manuscript describes the importance of sphingomyelin and cholesterol in preserving nuclear RNA. The authors profiled nuclear lipid microdomain RNA, and identified various mRNAs, lncRNAs, microRNAs and circular RNAs. They also found that sphingomyelinase treatment enhanced the effect of RNase treatment. All of the authors' conclusions are correlative and do not provide any strong conclusion. In order to develop this study further, the authors should provide experimental data supporting the consequence of RNA localization in the microdomains. Enhancement or inhibition of those RNA accumulation in the microdomains and the subsequent outcomes in cellular process (cell division cycle, proliferation, death, etc.) should be tested before publication. Molecular mechanisms of how those RNAs accumulate in the microdomain should be characterized even further.

Thank you very much for this observation. In the previous version, we demonstrated that increasing the concentration of SM-Chol in the medium, the formation of microdomains enriched in RNAs increases (Fig.5). We have now evaluated the cell morphology changes indicative of differentiation. The cells from their round shape acquire a differentiated phenotype (Fig.6). The specific markers for astrocytes (GFAP) and neurons (tubulin III) confirmed this result. It has been reported in the results (p.12) and specific methods (p. 24,25) were added. Understanding the exact molecular mechanisms will take a very long time. We are starting with a very complex and interdisciplinary study that will be the subject of a future article.

Reviewer #2 (Remarks to the Author):

In this manuscript entitled "Sphingomyelin and RNase-resistant RNA in Microdomains of the Inner Nuclear Membrane", Conte & Bulfoni et al., characterize the RNAs enriched in nuclear microdomains (NLMs) that are resistant to RNase treatment. Using the neuronal cell line HN9.10, the authors found that sphingomyelin and cholesterol protect certain mRNAs, miRNAs, and circRNAs against degradation. As a result, these RNAs became enriched in these nuclear sub-compartments and are believed to have gene expression regulatory functions important for neuronal health. This paper's findings are interesting, but it has several issues, related to experimental controls, data missing, and description of experimental design and results that need to be solved before

publication.

Besides this general comment, the paper will benefit from addressing the following concerns

1. The rationale beyond using the embryonic hippocampal cells needs to be explained because previous studies have been done in hepatocytes. It should be explained if the finding of RNAs in NMLs related to neuronal pathologies is a direct consequence of the cell line used in these studies.

It has been explained (p. 4 line 8 from the bottom and subsequent lines)

2. The statistical test, n, and significance should be provided for all figures.

It has been made

3. Figure 2 misses sufficient experimental details and quantification to support the conclusion. Only 2 cells are shown per analysis, and they look different from each other. There is no quantitative data for the immunofluorescences or indication of the number of cells or experiments done. The figure also misses a negative control and a marker for the cell plasma membrane. The scale bar should be added to the figure.

Figure was replaced (figure 4 in the present version). Lower magnification images with higher cell numbers have been reported. The percentage of positive cells was reported. A negative marker is really difficult to detect because all cells have sphingomyelin and cholesterol. The number of experiments done was reported in legend.

4. Figure 3, how does incubation with SM-Chol result in their increased content in the nucleus? This is not quantified or demonstrated, and it is needed to support the conclusions from these sections.

Specific experiments were added (Figure 5) – The effect of cell incubation with SM-Chol on the SM-Chol content in nuclear lipid domains was studied. It has been reported in the methods and in results (p.9 line 9 from the bottom and subsequent lines; p.27 line 9 from the bottom and subsequent lines).

5. Figure 4, in the text lines 233-235, Why the percentages of RNA classes do not sum to 100% and are different from the figure? What statistics were applied to support that there are differences between the nuclear and NML RNAs?

Thank you for your comment. The reported percentages represent the main RNA classes that differ between nuclei and NLMs. All subclasses are visible in Figure 9. Regarding protein-coding mRNA, miRNA, and miscRNA, a t-test was applied to verify that the detected differences were statistically significant. The corresponding *p-value* data have been incorporated into the text (p13 lines 1-9).

6. Relate with the conclusion from Figure 4 in which the authors found that mRNAs are less enriched in NMLs than in the nuclei. What is the rationale for analyzing mRNAs? And Why the number of enriched mRNAs in NMLs is similar to the number of microRNAs?

Previous research demonstrated that NLMs acted as rafts for anchoring transcriptionally active chromatin. So we decided to also analyze the mRNA (p. 14 and figure 9). Our results showed that NLMs are rich in miRNAs, so much so that they act at a similar value to mRNA (p. 20 lines 4-9). Furthermore, given that protein-coding RNAs, miRNAs, and miscRNAs (which may also include circRNAs) were the most differentially expressed and prominent RNA biotypes emerging from our analysis, we decided to further investigate them.

Since this was an exploratory analysis and a fishing expedition, we did not have prior expectations regarding the number of enriched miRNAs and mRNAs that would emerge from our study.

7. Figures 5 and 7, the findings by RNAseq should be validated by RT-qPCR. The meaning of the pools should be explained in the figure legend.

Thank you for your comment. We have validated the gene expression of 3 representative transcripts (*Ehmt2*, *Chaf1a* and *Dnm2*) in new and independent samples, confirming that the downregulation trend identified by RNA-seq is consistent with our expectations (p.14, lines 17,18). Additionally, we have removed the term "pool" and added the correct labels to the samples.

8. Figure 6, how many mRNAs are per Go-term category?

Thank you for your comment. The figure legend has been updated and expanded to include the number of mRNAs enriched in each GO-term category (Figure 10 in the present version).

9. In the sentence “The functional analysis of genes highly concentrated in the NLMs compared to the nuclei, which are 281 targeted by miRNAs at high concentrations in the NLMs compared to the nuclei.....” The meaning of targeted by miRNAs is not well explained. In the legend, there is a typo in “intensity”

The sentence has been revised (p.15 lines 6-10) as follows: "The functional analysis of genes highly concentrated in the NLMs compared to the nuclei, which are targeted by miRNAs highly concentrated in the NLMs compared to the nuclei according to associations from TarBase (citation: <https://pmc.ncbi.nlm.nih.gov/articles/PMC1370898/>), clearly indicates their involvement in chromatin remodeling and organization, as well as in the development of the nervous system (Figure 10)." Additionally, the typo in "intensity" has been corrected in the figure legend

10. This sentence needs references, probably a dot. “The association between these 307 miRNA and diseases was reported in Figure S5. It is evident that they are correlated with neuro psychological diseases as chronic epilepsy, amyotrophic lateral sclerosis and psychological disorders “.

The functional associations between miRNAs and diseases were predicted using Ingenuity Pathway Analysis (IPA). IPA is a widely used tool for analyzing omics data, enabling the identification of biological relationships, including disease associations, based on curated experimental evidence. All relevant information has been incorporated into the text (p.16 line 8 from the bottom and subsequent lines).

11. Data in Figure 8 need validation by RT-QPCR and demonstration that the RNAs are in fact circular RNAs.

Thank you for your comment. The identification of circRNAs was based on RNA-seq data and bioinformatic analysis, which predicted their circular nature. We then confirmed their expression in independent samples using RT-qPCR. However, we acknowledge that additional validation steps, such as RNase R treatment or the use of divergent primers, would be necessary to experimentally confirm their circularity (p.17 lines 2,1 from the bottom).

12. Discussion line 348. It will be helpful to have an explanation as for why the authors focused on mRNAs when circRNAs and miRNAs were previously found.

It has been reported (p.18, lines 2,1 from the bottom)

13. There is inconsistent formatting in the materials and methods section.

Material and methods section has been revised

Reviewer #3 (Remarks to the Author):

In their manuscript Conte et. al., Has come up with an idea of nuclear lipid microdomain with the inner nuclear membrane to host different classes of RNA - protected from RNases due to sphingomyelin. The biochemical isolation has been done to separate NLM-associated pool from total cellular pool. The authors have found candidate RNAs which are enriched for NLMs. This data have potential implications in physiological processes such as chromatin organization and brain disease pathology.

Although the hypothesis is interesting and potentially exciting to consider NLM as important sites for storage of miRNA and other RNAs for regulation of gene expression, the work is incomplete and misses several important controls that dampen my enthusiasm for this manuscript.

There are several issues that need attention:

The isolation of Nuclei and Nuclear associated microdomain should be confirmed by Western blot analysis for suitable marker proteins. This is important to conclude on the purity of the fractions they have studied in subsequent analysis and RNA estimation.

It has been made (results section p.5 lines 1-7, figure 1; materials and methods section p.26, line 4 from the bottom and subsequent lines)

It is essential also to perform super resolution microscopic analysis and TEM to comment on the size and shape of the microdomain with RNA within cells and after isolation as NLM. The microscopic data provided is not convincing for the reviewer.

Electron microscopy analysis has been performed. **(Results p.5 line 9 from the bottom and subsequent lines, figure 2; material and methods section p.24 lines 7-16)**

miRNAs are synthesized as pri- miRNAs and processed inside the nucleus to pre-miRNAs which are subsequently get exported to cytoplasm to get processed by Dicer1 in human cells cytoplasm to mature miRNA. I wonder what form of miRNAs (mature or pre-miRNA) that the authors detected with NLM. If I understand correctly, the mature miRNAs are followed. This is unusual as only a small fraction of miRNA is detected nucleus in normal condition. Therefore, if it is mature miRNA , then how they were getting retro-transported to nucleus from cytoplasm.

Thank you for your comment. Our miRNA-seq protocol is designed to capture small RNA molecules ranging from 18 to 30 nucleotides, which correspond to mature miRNAs, while pre-miRNAs are excluded during library preparation (size selection). Additionally, the mapping tool miRBase is optimized to detect mature miRNAs, further limiting the identification of precursor forms.

Although mature miRNAs are known for their cytoplasmic function, more evidence suggests they can undergo nuclear retro-transport through mechanisms involving Importin- α/β , Argonaute 2 (AGO2), and Exportin-1 (XPO1/CRM1). Within the nucleus, they participate in transcriptional regulation, RNA processing, and interactions with non-coding RNAs. Notably, exosomes may also contribute to intracellular miRNA trafficking, potentially facilitating the nuclear localization of specific miRNAs by interacting with nuclear lipid microdomains (NLMs). These findings indicate that nuclear miRNAs are functionally relevant and not merely residual cytoplasmic traces.

Our study found that the percentage of miRNAs in the nuclei is extremely low, while the highest proportion is observed in NLMs. We have incorporated this information into the Results (p.19 line 7 from the bottom and subsequent lines) and Discussion (p.16, lines 15-18) sections.

Schematic pictures depicting the research methodology and experimental flow should be included for better understanding the separation processes.

It has been included (Figure 13)

This email has been sent through the Springer Nature Tracking System NY-610A-NPG&MTS

Dear Editor,

Thank you very much for your message and suggestions of the referees that really improved the manuscripts. Accordingly, the manuscript has been revised.

I am sending you: 1) the manuscript with the previous revisions and with the new revisions; 2) the previous manuscript cleaned of revisions with only the new revisions highlighted

Reviewers' comments:

Reviewer #1 (Remarks to the Author):

The authors examined the morphological changes and differentiation phenotypes of HN9.10e cells. However, they still did not examine molecular mechanism of how exactly these events happen. Thus, this manuscript is still under the process of further development.

Possible mechanisms were reported in the discussion

Reviewer #2 (Remarks to the Author):

The authors have addressed the reviewers' comments; however, the quality of the experimental data should be improved. For example, the western blots in Figure 1 and the immunofluorescence images in Figure 4 appear overexposed, and the latter lacks a scale bar. It is also unclear whether the cells in the images are viable.

All images have been revised

Reviewer #3 (Remarks to the Author):

The authors have partly addressed my concerns in the revised manuscript. However, the manuscript still lacks any mechanistic information on the compartmentalization of RNAs to lipid-rich membrane microdomains of the nuclear membrane. I wish the authors would provide a hypothesis, at least in the discussion part, on how these mRNAs and miRNAs are getting associated with lipid-rich domains. These findings may have physiological importance and could thus be accepted for publication, but with additional textual changes introduced.

It has been included in the discussion